# Climatology of aerosol components concentration derived by GRASP algorithm from multi-angular polarimetric POLDER-3 observations

Lei Li[1], Yevgeny Derimian[2], Cheng Chen[2,3], Xindan Zhang[1], Huizheng Che[1*], Gregory L. Schuster[4], David Fuertes[3], Pavel Litvinov[3], Tatyana Lapyonok[2], Anton Lopatin[3], Christian Matar[3], Fabrice Ducos[2], Yana Karol[3], Benjamin Torres[2], Ke Gui[1], Yu Zheng[1], Yuanxin Liang[1], Yadong Lei[1], Jibiao Zhu[1], Lei Zhang[1], Junting Zhong[1], Xiaoye Zhang[1], Oleg Dubovik[2*]

[1]State Key Laboratory of Severe Weather (LASW) and Key Laboratory of Atmospheric Chemistry (LAC), Chinese Academy of Meteorological Sciences, CMA, Beijing, 100081, China
[2]Univ. Lille, CNRS, UMR 8518 - LOA - Laboratoire d'Optique Atmosphérique, F-59000 Lille, France
[3]GRASP-SAS, Villeneuve d'Ascq, France
[4]NASA Langley Research Center, Hampton, VA, USA

*Correspondence to*: Huizheng Che (chehz@cma.gov.cn) and Oleg Dubovik (oleg.dubovik@univ-lille.fr)

**Abstract.** The study presents a climatology of aerosol composition concentration obtained by a recently
developed GRASP/Component (GRASP stands for the Generalized Retrieval of Atmosphere and Surface Properties algorithm) approach applied to the whole archive of POLDER-3 observations. The conceptual specific of the GRASP/Component approach is in direct retrieval of aerosol speciation (component fraction) without an intermediate retrievals of aerosol optical characteristics. Despite a global validation of the derived aerosol component product is challenging, the results obtained are in line with general
knowledge about aerosol types in different regions. In addition, we compare the GRASP derived black carbon and dust components with those of the MERRA-2 (Modern-Era Retrospective Analysis for



Research and Applications, version 2) product. Quite reasonable general agreement in spatial and temporal distribution of the species provided by GRASP and MERRA-2 was found, the differences however appeared in regions known for strong biomass burning and dust emissions; the reasons for the

discrepancies are discussed. The other derived components, such as concentration of absorbing (black carbon, brown carbon, iron oxides content in mineral dust) and scattering (ammonium sulphate and nitrate, organic carbon, non-absorbing dust) aerosol, represent scarce but imperative information for validation and potential adjustment of chemical transport models. The aerosol optical properties derived by GRASP/Component were found to agree well with the AERONET ground reference data and fully

consistent with the previous GRASP Optimized, High Precision and Models retrieval versions applied to POLDER-3 data. Thus, the presented extensive climatology product provides an opportunity for understanding variabilities and trends in global and regional distributions of aerosol species. The climatology of the aerosol components obtained in addition to the aerosol optical properties provides additional valuable, qualitatively new inside about aerosol distributions and, therefore, demonstrates

advantages of multi-angular polarimetric satellite observations as the next frontier for aerosol inversion from advanced satellite observations. The GRASP/Component products are publicly available (https://www.grasp-open.com/products/, last access: 15 March 2022) and the dataset used in the current study is registedred under https://doi.org/10.5281/zenodo.6395384 (Li et al., 2022).

## 1 Introduction

The latest IPCC have reported that aerosols are still the most contributor to the large forcing uncertainty (Forster et al., 2021) with the assessment of -2.0 and -0.4 W/m$^2$ (90% likelihood) for the aerosol effective radiative forcing (Bellouin et al., 2020). One of the main factors responsible for the large aerosol radiative forcing uncertainty is the lack of understanding and knowledge on global aerosol composition distribution. The potential of using remote sensing observations has already demonstrated in several studies. For

example, Dubovik et al. (2002b) discussed the differences of spectral absorption between carbonaceous particles and mineral dust using retrieval from the global AERONET (Aerosol Robotic Network) measurements. Schuster et al. (2005), for the first time, demonstrated a potential to infer the black carbon



(BC) content from remote sensing AERONET data using an assumption that all aerosol absorption could be interpreted to the contribution of BC component. Then, the hematite and brown carbon (BrC)

proportions were successfully estimated by Koven and Fung (2006) and Arola et al. (2011), respectively, from AERONET products using the spectral variability of themselves complex refraction indices. Derimian et al. (2008) discussed the difficulties in the separation of carbonaceous species absorption from mineral dust (largely iron oxides) absorption and highlighted the importance of having sufficient spectral resolution of measurements to capture the differences in absorption spectral dependence among BC, BrC,

and iron oxides are essential for determination of BC, BrC, and iron oxides proportions. With additional constraints on spectral single scattering albedo and the complex refractive index (Arola et al., 2011; Koven and Fung, 2006; Schuster et al., 2005), the proportions of BC, BrC and dust can be estimated simultaneously based on AERONET products (Wang et al., 2013). Recently, Schuster et al. (2016) successfully retrieved the proportions of aerosol absorbing components (BC, BrC, and iron oxides) in

fine- and coarse-mode particles of biomass burning and dust aerosols over the globe based on the aerosol products (aerosol size distributions and complex refraction index) provided in the AERONET retrievals described by Dubovik and King (2000) and Dubovik et al. (2002b, 2006). The estimation of aerosol compositional content based on AERONET optical retrievals are also discussed in several other studies (Bahadur et al., 2012; Cazorla et al., 2013; Chung et al., 2012; Costabile et al., 2013; Dey et al., 2006; Li

et al., 2013, 2015; Russell et al., 2010; Xie et al., 2014, 2017; Zhang et al., 2018). AERONET products (such as particle size distribution, wavelength dependence of extinction, single scattering albedo) are also used to distinguish the aerosol types at several regional sites (Giles et al., 2012; Logan et al., 2013). The AERONET-derived aerosol properties together with the active measurements of lidar ratio and spectral depolarization ratio also were used by Burton et al. (2012, 2014) for aerosol types classification from the

airborne High Spectral Resolution Lidar (HSRL) measurements. Ganguly et al. (2009b, 2009a) retrieved the concentration of several aerosol components from the combination of AERONET sun photometer and lidar measurements. It is noted that the evaluation of aerosol components by all above approaches are conducted using the remote sensing measurements at several discrete sites even for the globally distributed ground-based AERONET sites.






Satellite remote sensing was extensively used in diverse efforts for improving global aerosol monitoring: for characterizing spatial aerosol distribution (Kaufman et al., 2002; Remer et al., 2008), data assimilation (Dubovik et al., 2008; Chen et al., 2018, 2019) and modeling of aerosol climate effects (Bellouin et al., 2005; Jia et al., 2021; Myhre, 2009; Yu et al., 2006), etc.. A variety of inversion algorithms have been

proposed and applied to different and sometimes the same space-borne instruments for further understanding of the atmospheric aerosol properties on large spatial scale (Bréon et al., 2011; Dubovik et al., 2019; King et al., 1999; Li et al., 2009). For example, Dark Target (DT) (Levy et al., 2013; Remer et al., 2005, 2020), Deep Blue (DB) (Hsu et al., 2006, 2013), and Multi-Angle Implementation of Atmospheric Correction (MAIAC) (Lyapustin et al., 2018) are proposed for inverting the Moderate

Resolution Imaging Spectroradiometer (MODIS) observations. All these algorithms enforced the MODIS retrieval and made MODIS aerosol data base a reference aerosol product in the community. At the same time, MODIS a single-view instrument and generally does not provide sufficient information to identify aerosol types. In contrast, the multi-angular intensity observations by MISR (Multiangle Imaging Spectroradiometer) have been used successfully to identify aerosol type in several regional-scale studies

including wildfire smoke (Chen et al., 2008; Guo et al., 2013), desert dust (Guo et al., 2013; Kahn et al., 2009), volcanic ash (Kahn and Limbacher, 2012; Scollo et al., 2012). Nonetheless, although the aerosol classification from MISR data sets has been supported by above mentioned studies these studies provided mainly qualitative flagging of aerosol types for observed aerosol events. In addition, because of the aerosol types defined by aerosol parameters in the look-up table (LUT), the best fit to the observations

for intermediate retrievals were obtained using discrete set of possible solutions. In this regards, the numerous studies suggested that multi-angle polarization measurements has very high information content and are most appropriate for providing potentially for advanced aerosol product enhanced inside about aerosol microphysics and composition (Hasekamp and Landgraf, 2007; Knobelspiesse et al., 2011; Mishchenko and Geogdzhayev, 2007; Mishchenko and Travis, 1997; Tanré et al., 2011). However, due

to rather limited amount of available multi-angle polarization observations and the complexity in their interpretation, the impact of satellite polarimetry on aerosol monitoring remains fairly. Nonetheless, as discussed by Dubovik et al. (2019, 2021b) there is large number of multi-angle polarimeter missions is planned and significant progress has been achieved in development of the algorithms of new generation




for interpretation polarimetry observation. Therefore, the impact of polarimetric data on overall aerosol
science is expected to increase rapidly in coming years.

The POLDER (PoLarization and Directionality of the Earth's Reflectances) instrument was a satellite
polarimeter designed for collecting the spectral directional polarized solar radiation measurements
(Deschamps et al., 1994). POLDER-3, which was launched onboard PARASOL (Polarization &
Anisotropy of Reflectances for Atmospheric Sciences coupled with Observations from a Lidar), provided
the longest records of multi-angle polarized measurements from March 2005 to October 2013. In order
to invert the POLDER multi-angle polarized measurements, several algorithms following the idea of a
conventional MODIS-like look-up-table (LUT) approach (Kaufman et al., 1997; Tanré et al., 1997) were
originally proposed to retrieve aerosol optical depth (AOD) product over ocean (Deuzé et al., 2001;
Goloub et al., 1999; Herman et al., 2005) and fine-mode AOD (AODF) over land (Deuzé et al., 2001;
Herman et al., 2005). However, the exploration of the full extent of the aerosol information contained in
MAP measurements, start to be feasible only relatively recently with the development of the so-called
satellite "algorithms of new generation", such as the SRON (Fu and Hasekamp, 2018; Hasekamp et al.,
2011) and GRASP (Dubovik et al., 2011, 2021a) retrievals. These algorithms consider a continuous space
of aerosol microphysical properties (size distribution, refractive index), instead of using standard aerosol
models, and retrieve the parameters of underlying surface simultaneously with aerosol properties to
properly account for land or ocean reflection. Specifically, the data analysed in this paper were generated
in frame of algorithm named GRASP (Generalized Retrieval of Atmosphere and Surface Properties), that
was initially developed for enhanced retrieval of aerosol properties from POLDER observations (Dubovik
et al., 2011) and then the applicability of GRASP has extended to wide variety of diverse remote sensing
applications including both passive and active observations (see details in the papers by Dubovik et al.,
2014, 2021a). After about a decade of development and advancement, GRASP algorithm has been
adapted for operational generation extended aerosol products from POLDER observations. Several data
sets of enhance aerosol retrieval from POLDER observations (see Section 4 in Dubovik et al., 2021a) that
now are publicly available (archived at the AERIS/ICARE Data and Services Center
(http://www.icare.univ-lille1.fr) and GRASP-OPEN site (https://www.grasp-open.com). All



POLDER/GRASP retrieval releases include extended set of conventional aerosol optical parameters (such as spectral aerosol optical depth, spectral aerosol absorption optical depth, spectral fine-mode aerosol optical depth, spectral coarse-mode aerosol optical depth, particle size distribution, single-scattering albedo, complex refractive index, fraction of spherical particles, etc.) (Chen et al., 2020). In addition, the latest POLDER-3/GRASP-Component processing provided also some information of aerosol composition by estimation of fractions of the such aerosol components as black carbon, brown carbon, absorbing and scattering mineral dust, etc. (Li et al., 2019).

This GRASP/Component approach derives fractions of aerosol components together with size distribution and non-spherical fraction of aerosol particle directly from the measured radiances without an intermediate step of optical aerosol properties retrieval. This is significant methodological difference with other approaches for aerosol typing from satellite observations. For example, Russell et al., (2014) also demonstrated a possibility to identify aerosol type from POLDER-3 observations. However, that study analyzed aerosol optical properties derived from POLDER-3 by SRON algorithm using of aerosol classes/types (pure dust, polluted dust, biomass burning, urban/industrial, and marine aerosol) set of criteria formulated based on the AERONET retrievals (e.g., aerosol optical and microphysical parameters). In should be noted also, that GRASP/Component approach was applied to AERONET data and was demonstrated to derive valuable information about aerosol components , as well as to provide optical properties with the accuracy comparable to the standard AERONET retrievals (Li et al., 2019; Zhang et al., 2022).

Because the validation of aerosol optical properties (AOD, AE, AODF, AODC, AAOD, SSA, etc.) generated by POLDER-3/GRASP-Component approach has been realized and discussed by Zhang et al. (2021). This study presents the extensive analysis of aerosol composition products generated by GRASP/Component from full archive of the multi-angular polarimetric POLDER-3 observations. We present the climatological, global statistical analysis of satellite-retrieved aerosol absorbing components and scattering components with the assessment of consistency with general expectation to the extent possible. We also make comparisons of GRASP/Component BC and dust retrievals to those in MERRA-



2 (Modern-Era Retrospective Analysis for Research and Applications, version 2) products. Our expectation, based on preliminary sparse analysis of derived aerosol component information and on the results of study by Zhang et al. (2021b), is that GRASP/Component approach generally can provide from POLDER like polarimetric data both conventional aerosol optical parameters and the parameters characterizing aerosol composition with sufficiently high accuracy.

## 170  2 Data and methodology description

### 2.1 POLDER-3 sensor

The PARASOL was launched on December 18, 2004 as a part of the A-Train constellation. POLDER-3, designed for the measurements of multispectral, multidirectional, and polarized radiances, onboard PARASOL satellite provides data for the period of March 2005 – October 2013 (Tanré et al., 2011).

POLDER-3 has nine channels (440, 490, 565, 670, 763, 765, 865, 910, and 1020 nm) for the measurements of total radiance and three channels (490, 670, and 865 nm) for the measurements of polarization. The measurements are collected in up to 16 viewing angles per pixel with a resolution of 5.3 × 6.2 km$^2$ at nadir. In addition, it has been demonstrated that the multipolarization and multidirectional measurements of POLDER help to distinguish better cloudy and clear pixels (Buriez et al., 1997; Goloub

et al., 1997; Parol et al., 1999; Zeng et al., 2011).

### 2.2 GRASP/Component product

We are discussing the advanced aerosol products obtained from POLDER-3 using GRASP algorithm. The efficiency and flexibility of GRASP has been illustrated in many previous studies on applications to diverse remote sensing instruments, such as ground-based AERONET photometers and lidars (e.g.,

Benavent-Oltra et al., 2019; Lopatin et al., 2013, 2021; Titos et al., 2019; Tsekeri et al., 2017; Zhang et al., 2022, etc.), POLDER-3 (e.g., Chen et al., 2020; Dubovik et al., 2011; Li et al., 2019, 2020b, 2020a, etc.), DPC/GF-5 (Li et al., 2022), sky cameras (e.g., Román et al., 2017, 2021, etc.), polar-nephelometer data (e.g., Espinosa et al., 2017; Schuster et al., 2019, etc.). The papers by Dubovik et al. (2011, 2014, 2021a) presents the detailed description of the main concept of GRASP as well as the specific





methodological introductions. In brief, GRASP employs the multi-term least square minimization
(Dubovik, 2004; Dubovik et al., 2021a) for the implementation of statistically optimized fitting. The
advantages of this concept have been initially demonstrated in the retrievals of the aerosol properties from
ground-based AERONET sun/sky-radiometers (Dubovik et al., 2000, 2002a, 2002b, 2006; Dubovik and
King, 2000). One of essential differences from traditional LUT algorithms is that GRASP performs online

radiative transfer computations for multiple interactions of the scattered solar light in the atmosphere.
That is, GRASP searches the optimized fitting in the continuous space of solutions, whereas the traditional
LUTs approaches search among the pre-computed solutions. This allows GRASP to explore a much larger
parameter space (i.e., non-lognormal size distributions and orders of magnitude refractive indices) than
tratidional approaches. The properties of surface reflectance in GRASP are modelled using Bidirectional

Reflectance Distribution Function (BRDF) and Bidirectional Polarization Distribution Function (BPDF)
models. Specifically, over land, the Ross-Li BRDF (Li and Strahler, 1992; Ross, 1981) and BPDF
(Maignan et al., 2009) models are used considering geometrical and volumetric terms of the Ross-Li
BRDF models nearly spectrally constant (Litvinov et al., 2011). The reflective properties of ocean surface
are taken into account using Cox-Munk model (Cox and Munk, 1954) analogously to the many

conventional approaches, the details are provided by Dubovik et al. (2011) and Frouin et al. (2019). The
GRASP is open-source code that is provided together with a described documentation at GRASP-OPEN
website https://www.grasp-open.com (last access: 30 November 2021).

Several different versions of aerosol optical products derived by previous GRASP (Optimized, High

Precision, and Models) approaches for the full archive of POLDER-3 observations have been generated
and released for open access (Dubovik et al., 2019; Chen et al., 2020). Here, we provide only a brief
description of these three approaches and their main differences that are relevant to present analysis. As
described in Dubovik et al. (2011), the solution vector in the all these three GRASP configurations can
be written as Eq. (1):

$\mathbf{a} = (\boldsymbol{a}_v, \boldsymbol{a}_n, \boldsymbol{a}_k, \boldsymbol{a}_{sph}, \boldsymbol{a}_{Vc}, \boldsymbol{a}_h, \boldsymbol{a}_{brdf1}, \boldsymbol{a}_{brdf2}, \boldsymbol{a}_{brdf3}, \boldsymbol{a}_{bpdf})^T,$                    (1)

where the elements representing the normalized logarithms of $dV(r)/dlnr$, real part of complex
refractive index, imaginary part of complex refractive index, spherical particles fraction, total volume





concentration, mean altitude of the aerosol layer, the parameters in BRDF and BPDF model are characterized by the vectors of $\boldsymbol{a}_v$, $\boldsymbol{a}_n$, $\boldsymbol{a}_k$, $\boldsymbol{a}_{sph}$, $\boldsymbol{a}_{Vc}$, $\boldsymbol{a}_h$, $\boldsymbol{a}_{brdf1}$, $\boldsymbol{a}_{brdf2}$, $\boldsymbol{a}_{brdf3}$, $\boldsymbol{a}_{bpdf}$, respectively.

The GRASP aerosol optical products derived by Optimized, HP and Models approaches are generated and released publicly as daily, monthly, seasonal, yearly and climatological datasets in Level-2 and Level-3 (Chen et al., 2020). Level-2 represents the quality-filtered products with an initial resolution of satellite observations. Level-3 represents the products with a resolution of 0.1° and 1° grid box, respectively. Hereafter all the GRASP/Component climatology is presented at 0.1° × 0.1° spatial resolution, except,

the comparisons with MERRA-2 will be at 0.5° × 0.625°. Level-0 represents the raw data and Level-1 represents daily files of retrievals considered as results of intermediate processing, which could be provided under request.

As mentioned above, GRASP is a versatile algorithm, that can be applied to diverse observations and can

be served as an assessable community tool for testing new and innovative ideas for interpretation of remote sensing measurements, e.g. using synergy of different observations (Dubovik et al., 2021b; Lopatin et al., 2021). The most recent "GRASP/Component" approach described in the study of Li et al. (2019), can derive some size-resolved aerosol composition information attempting to discriminate the contribution of such species as black carbon, brown carbon, coarse-mode absorbing (mainly representing

iron oxides in mineral dust), fine- and coarse-mode non-absorbing soluble and insoluble, etc. Specifically, forward model of GRASP/Component uses the fixed refractive index for each component and based on the given component fractions calculates the complex refraction indices of aerosol mixture for simulating the observations. Correspondingly, the state vector in the GRASP/Component configuration includes the component factions instead of refractive indices (Li et al., 2019):

$\mathbf{a} = (\boldsymbol{a}_v, \boldsymbol{a}_{fract}, \boldsymbol{a}_{sph}, \boldsymbol{a}_{Vc}, \boldsymbol{a}_h, \boldsymbol{a}_{brdf1}, \boldsymbol{a}_{brdf2}, \boldsymbol{a}_{brdf3}, \boldsymbol{a}_{bpdf})^T,$  (2)

where $\boldsymbol{a}_{fract}$ represents the aerosol component fractions.

Thus, the main conceptual difference of GRASP/Component from GRASP Optimized and High Precision is the retrieval of fractions of 6 components instead of the real and imaginary parts of complex refraction

index retrieval for each wavelength (12 parameters in GRASP Optimized and High Precision). Thus, the



number of unknowns is reduced in GRASP/Component compared to GRASP Optimized, High Precision. In addition, the spectral refractive index of each species assumed in the component model presents extra constraints on the spectral dependences of complex refraction index of aerosol mixture. The advantage in GRASP/Component associated with the reduction of unknowns and additional constraints on spectral

dependences was discussed by the study of Zhang et al. (2021). In the contrast, the GRASP/Models approach retrieves even smaller number of parameters since it considers aerosol as an external mixture of aerosol components with a priori fixed all optical properties (particle size, refractive index and fraction of spheres). However, GRASP/Models seem to have some limitations in deriving such detailed parameters as AE, AODC, AODF, etc. (Chen et al., 2020).


The GRASP/Component approach assumes internal mixing of aerosol component for fine and coarse aerosol modes, which requires a mixing rule on the estimation of components for the aerosol mixture. Maxwell-Garnett (MG) effective medium approximation is employed in GRASP/Component, as one of main mixing rules, to estimate an effective refractive index of aerosol mixture considering several

insoluble components presenting soluble host, such as ammonium nitrate (or sulphate) (Bohren and Huffman, 1983; Lesins et al., 2002). The hygroscopicity of ammonium nitrate has been quantitatively described in details (Tang, 1996), thus the proportions and properties of ammonium nitrate and aerosol water content are selected for calculation of optical properties of the host. Thus, using fractions of the components and relative humidity as variable parameters, the refractive index of a particle composed by

several insoluble components (e.g., BC, BrC, mineral dust etc.) suspended in such host were determined by the MG equations based on the calculation of electric fields.

Currently, the complete POLDER-3 data archive has been processed by GRASP/Component approach to provide extensive aerosol component concentration and aerosol optical properties retrievals. Similar to

previous products described in Chen et al. (2020), extensive GRASP/Component products generated as daily, monthly, seasonal, yearly and climatological datasets in Level-2 and Level-3 are publicly assessable at GRASP-open (https://www.grasp-open.com/). Although several regional applications of this GRASP/Component have been discussed in several studies (Li et al., 2020b, 2020a; Zhang et al., 2021),





the climatology of derived aerosol component distribution has not yet been fully analyzed in details and

therefore the full potential of this new satellite derived aerosol composition was not fully clarified for the broad aerosol community. Therefore, the main focus of the study in paper is on analysis/verifications of aerosol composition (fraction) retrievals and considering the apparent of aerosol composition climatological patterns. Indeed, Zhang et al. (2021) validated GRASP/Component optical properties against AERONET data and concluded that generated total AOD values have minimal bias both over land

and ocean similar to total AOD provided by GRASP/Models, while the detailed properties such as AE, AODF and AODC have similarly good validation metric as GRASP/HP. This suggested that GRASP/Component provided the overall most consistent both total and detailed aerosol properties.

### 2.3 MERRA-2 data

MERRA-2 is the new version of atmospheric reanalysis publicly distributed by the NASA Global

Modeling and Assimilation Office (Gelaro et al., 2017; Randles et al., 2017) and the detailed description can be found in (Buchard et al., 2017; Gelaro et al., 2017; Randles et al., 2017). The AOD products derived from the ground-based AERONET measurements and satellite observations (e.g., MODIS, MISR) are assimilated into the MERRA-2 by the GEOS-5 (Goddard Earth Observing System). The GOCART (Goddard Chemistry, Aerosol, Radiation and Transport) model is employed for the simulation of aerosol

compositions such as black carbon, organic carbon, dust, etc. Up to now, numerous studies presented quantificational validations and assessments of different MERRA-2 products on a global or regional scale using measurements, such as AOD (Randles et al., 2017; Song et al., 2018; Sun et al., 2019), $PM_{2.5}$ (Buchard et al., 2016; Song et al., 2018), and surface BC concentration using measurements at several separate sites in China (Qin et al., 2019; Xu et al., 2020). However, MERRA-2 BC and dust columnar

concentration products have not been compared to any observations globally. In this study, we employ the monthly MERRA-2 BC and dust columnar concentration products in a spatial resolution of 0.5° × 0.625° as available from the NASA website (https://daac.gsfc.nasa.gov).



# 3 Global POLDER-3 GRASP/Component aerosol products

The POLDER-3 GRASP/Component retrieval provides following parameters characterizing the aerosol composition: volume fractions of Black Carbon, Brown Carbon, fine/coarse mode non-absorbing soluble and insoluble, coarse mode absorbing, as well as fine/coarse mode aerosol water content. From the view point of optical properties, the strongly absorbing and mostly scattering aerosol components are convenient to analyse separately.

## 3.1 GRASP/POLDER-3 absorbing component products

The following aerosol components determine the aerosol absorption: BC and BrC in fine mode particles and coarse mode absorbing insoluble (CAI) component mainly representing iron oxides element contained in mineral dust.

### 3.1.1 Black Carbon (BC)

Figure 1 presents the distribution of seasonal BC columnar mass concentration in the climatological products derived by the GRASP/Component from POLDER-3 observations and Figure 2 shows the corresponding standard deviations (STD) divided by the mean of BC concentration. These satellite-derived BC concentrations present a noticeable temporal and spatial variations that are consistent with the distribution of biomass burning events. For example, over the African continent, the high BC concentrations can be observed over the Sahel region during the DJF season extending to southern Africa region during the JJA and SON seasons as intense agricultural burnings occurring in the sub-Sahelian region of Africa and progressing from north to south following the African monsoon cycle. In addition, large STD/Mean values are obtained for BC concentrations over southern Africa region during the SON season. The elevated BC concentrations are also retrieved over South America during the entire biomass burning season (from August to November, see Fig. 1), which is in line with the previous studies (Koren et al., 2007; Lei et al., 2021). At the same time, high STD/Mean values for BC concentration are obtained during the SON season over the South America, which indicates the large interannual variability. Natural





factors such as weather and cultural practices are known to make important contributions to the significant interannual variability of biomass burning in the Amazon basin rainforest (Koren et al., 2007). Fig. 2 also

indicates that low STD/Mean values for BC concentration in Asia, Europe and North America are observed when the intensity of anthropogenic emissions is small. Yearly means of BC columnar mass concentration for the period from March 2005 to October 2013 is presented in Fig. 3. We should point out that many BC particles are generated from anthropogenic activities such as in China and India (shown in Fig. 4), however, our global climatology also indicates that the BC concentration emitted from biomass

burning in Africa and South America produces much higher BC concentrations than the anthropogenic emissions, e.g., in China and India (see Fig. 1 and 3). The climatology of the BC mass concentrations over East and South Asia regions are studied in details by Li et al. (2020c). A reasonable agreement between in situ ground-measured and satellite-derived BC mass concentration in China has been early demonstrated by Li et al. (2020c). For instance, the BC product of GRASP/Component and the in situ

measurements agrees with the mean absolute difference of about 2.7 $\mu m/m^3$, ranging from 0.09 $\mu m/m^3$ to 7.8 $\mu m/m^3$ and the relative difference of about 40%, ranging from 2.6% to 60%, respectively (Li et al., 2020b). High BC proportions in Asia are found to be associated strongly with biomass burning events (such as agriculture waste open burning event during MAM over the Indo-China Peninsula or JJA in the North China Plain) and fossil fuel combustion (Li et al., 2020b). It is noted that high BC concentration in

India and China during the DJF season is strongly associated with anthropogenic activity (e.g., Li et al., 2019). The interpretation of some BC concentration over ocean, especially in high latitude near the polar region, should be taken with more caution because the sensitivity to the absorption and thus to the BC signal is limited when the aerosol loadings are very low and the retrievals are more uncertain, which is the same situation for large STD/Mean values when BC concentration is very low. In this respect it can

be noted that based on the sensitivity tests and uncertainty assessments in the study of Li et al. (2019), the uncertainty in BC fraction is within 50% when AOD (440 nm) is larger than 0.4 and fraction is higher than 0.01. This BC uncertainty is mainly resulting from the reported in the literature highly variable complex refractive index of BC, e.g., from 1.75 + 0.63i to 1.95 + 0.79i (Bond et al., 2013; Bond and Bergstrom, 2006). Similarly, the uncertainty in BC retrievals from ground-based AERONET

measurements is about 50% (Schuster et al., 2016b).

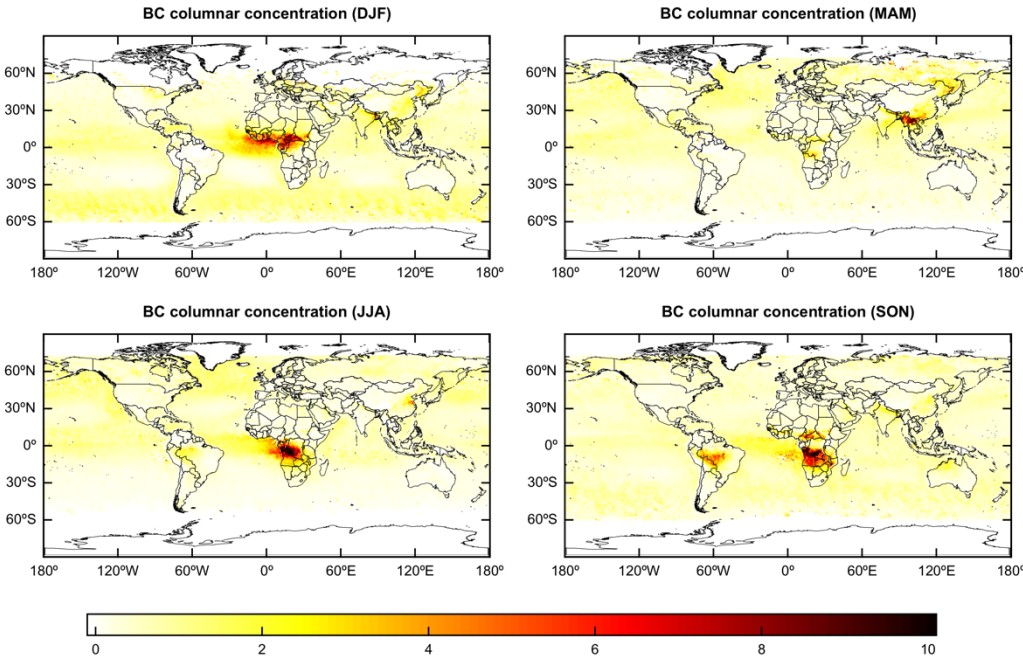

**Figure 1:** Spatial distribution of seasonal BC columnar mass concentration (mg/m$^2$) derived by the GRASP/Component approach from POLDER-3 observations. DJF: December-January-February; MAM: March-April-May; JJA: June-July-August; SON: September-October-November.


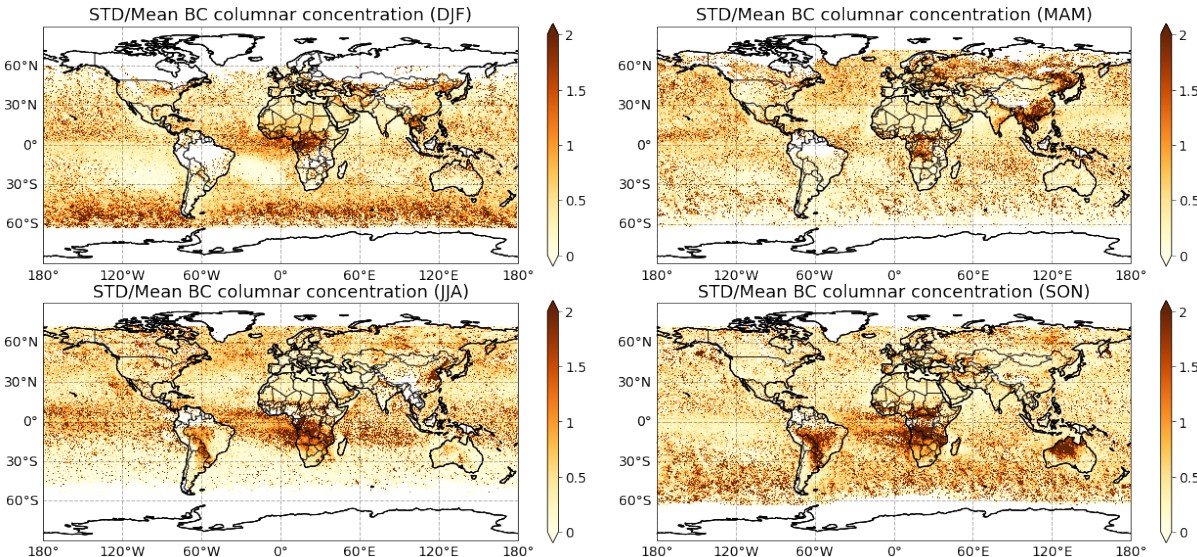

**Figure 2:** The standard deviation (STD)/Mean for seasonal BC columnar mass concentration in Fig. 1 derived by the GRASP/Component approach from POLDER-3 observations. DJF: December-January-February; MAM: March-April-May; JJA: June-July-August; SON: September-October-November.






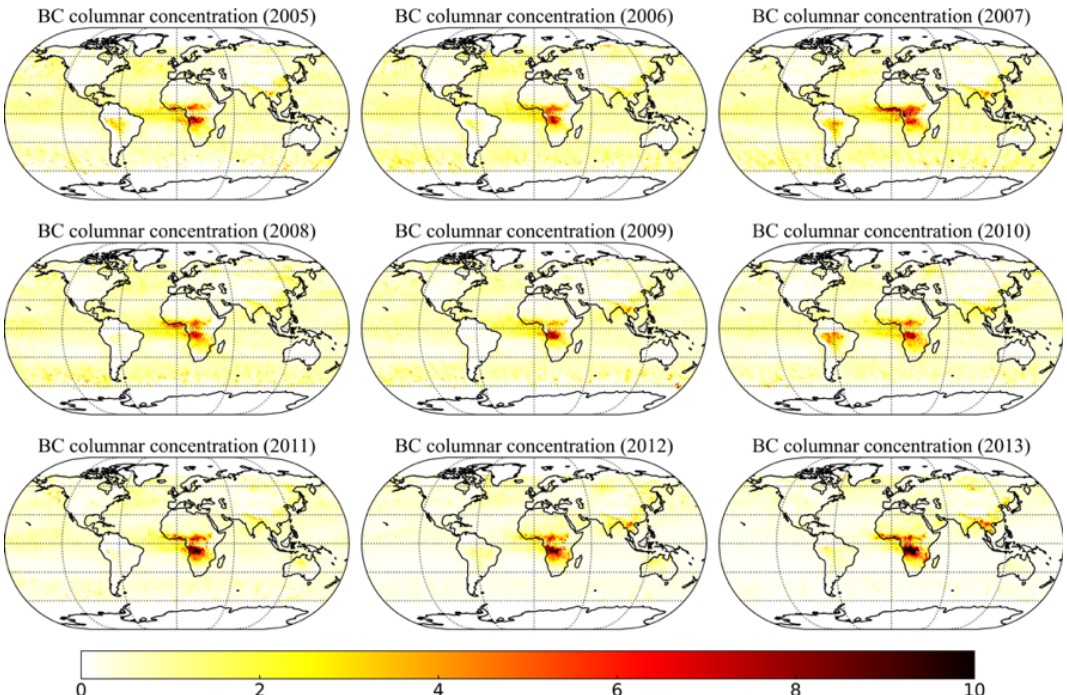

**Figure 3:** Yearly mean BC columnar mass concentration (mg/m$^2$) in resolution of derived by GRASP/Component approach from POLDER-3 observations for the period from March 2005 to October 2013.

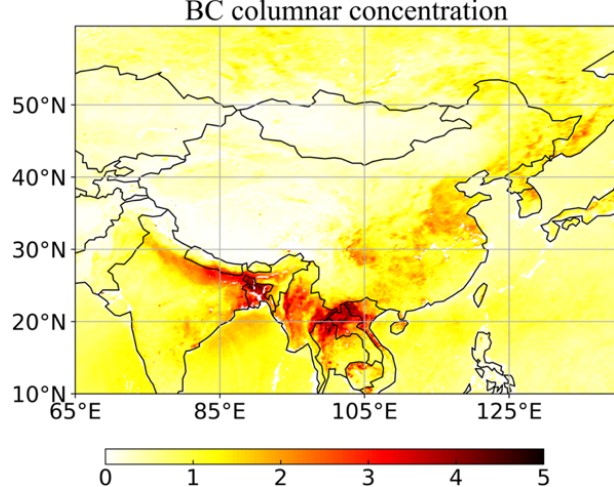

**Figure 4:** Climatological BC columnar mass concentration (mg/m$^2$) derived by the GRASP/Component approach from POLDER-3 observations for the period from March 2005 to October 2013 in Asia.






### 3.1.2 Brown Carbon (BrC)

Unlike strong and spectrally flat absorption of BC component, BrC shows a strong spectral dependence of absorption that decreases from shorter to longer wavelengths and a weaker absorption than BC (Chakrabarty et al., 2010; Kirchstetter et al., 2004; Wonaschütz et al., 2009). Fig. 5 and 6 present the

satellite-derived seasonal BrC columnar mass concentration issued by GRASP/Component for the period 2005－2013 and the corresponding standard deviations (STD) divided by the mean of BrC concentration, respectively. The seasonal variation and spatial distribution of BrC columnar mass concentration show similarities to the BC pattern (in Fig. 1) as these light-absorbing carbonaceous particles are generally emitted from same emission sources such as biomass or fossil fuel combustion. Therefore, high BrC

concentration are also observed over Africa, South America and the Indo-China Peninsula in Asia during the corresponding biomass burning period as discussed above for the BC concentration. High BrC concentration observed over the ocean near the continent is rather associated with the aerosol transport. For example, strong fires occurring from north to south Africa during the whole biomass burning period make large contributions to the BrC particles burden over the Atlantic Ocean near the African continent

during the DJF, JJA, and SON seasons. Elevated BrC concentrations are also seen over the Pacific Ocean during the MAM season that can be transport of biomass burning emissions from the Indo-China Peninsula (Li et al., 2020b). The observed in the global inter-annual climatology (Fig. 1 and Fig. 5) seasonal variabilities of BC and BrC concentrations in Asia were already analysed in the earlier study by Li et al. (2020c). The elevated BC concentrations in this work were attributed to fresh biomass burning

aerosols while those of the BrC to the aged biomass burning aerosols. Indeed, interactions with atmospheric gases (e.g., $O_3$, $NO_2$, and $SO_2$) together with higher atmospheric water content over ocean can change the morphology and optical property of BC particles by oxidization and coating, e.g., (Adachi and Buseck, 2013; China et al., 2013; Decesari et al., 2002; Zhang et al., 2008). Therefore, the meteorological factors and the atmospheric gases variability are also expected to contribute to the BrC

seasonal and interannual variability. It can be seen from Fig. 6 that STD/Mean values present no significant differences, which is same to the small interannual variability of BrC concentration in Fig. 7

Earth System
Science
Data

except over the southern Africa during the SON season (low concentration in 2012 and 2013). For the proper data interpretation, it is worth noting that the uncertainty in BrC fraction is normally less than 50% if the BrC fraction is above 0.1, even for very low AOD (smaller than 0.05), although the uncertainty is large in the case of small BrC fraction and low aerosol loading; the error estimations for GRASP/Component were conducted in (Li et al., 2019).

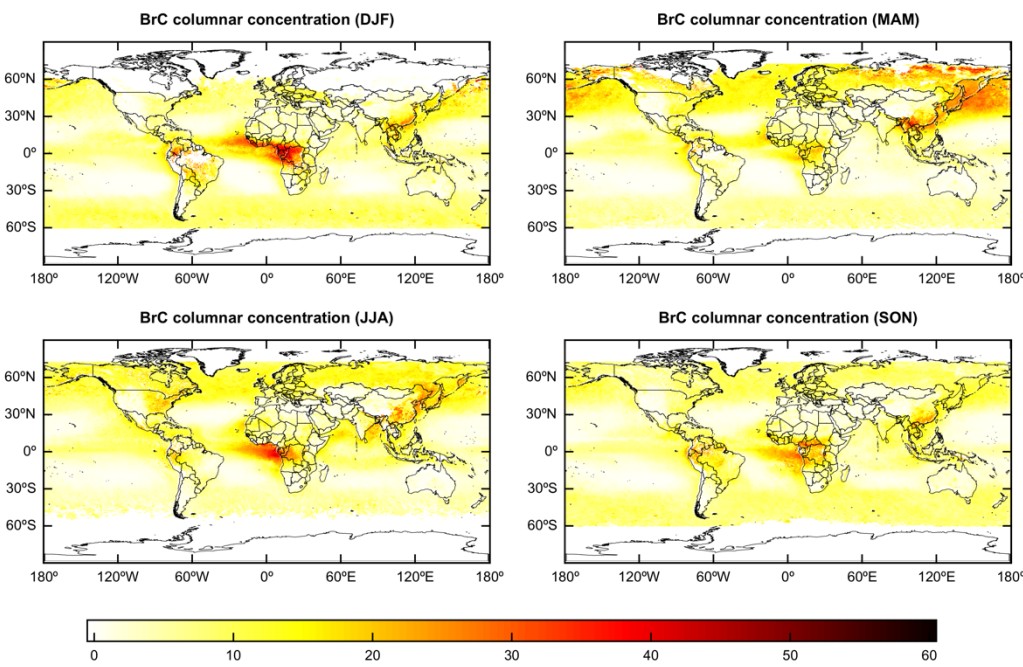

**Figure 5:** Spatial distribution of seasonal BrC columnar mass concentration (mg/m$^2$) derived by the GRASP/Component approach from POLDER-3 observations. DJF: December-January-February; MAM: March-April-May; JJA: June-July-August; SON: September-October-November.

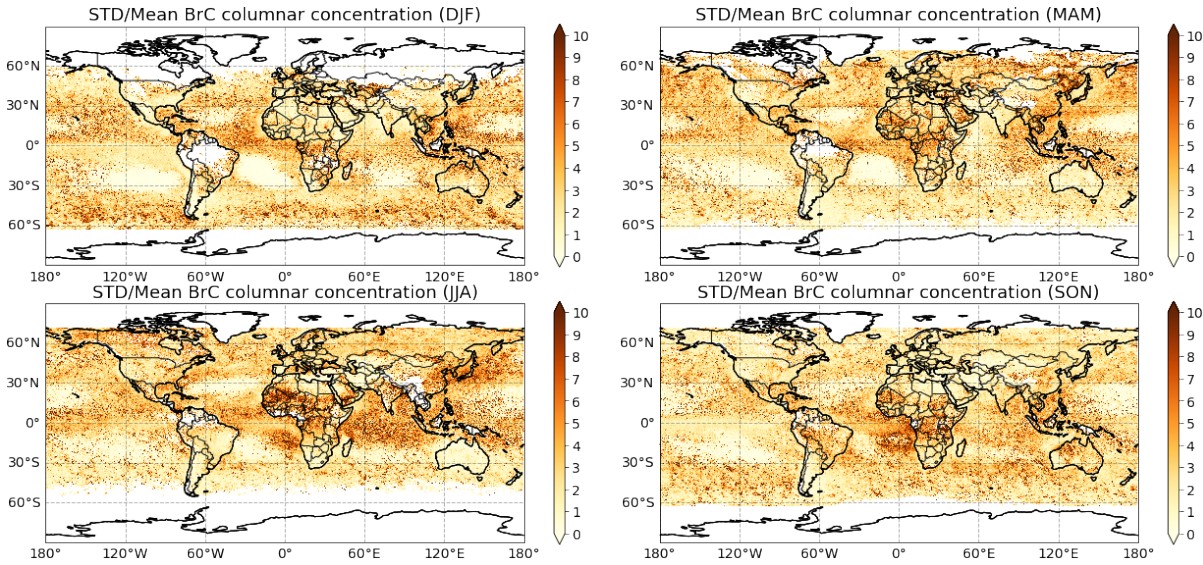

**Figure 6:** The standard deviation (STD)/Mean for seasonal BrC columnar mass concentration in Fig. 5 derived by the GRASP/Component approach from POLDER-3 observations. DJF: December-January-February; MAM: March-April-May; JJA: June-July-August; SON: September-October-November.


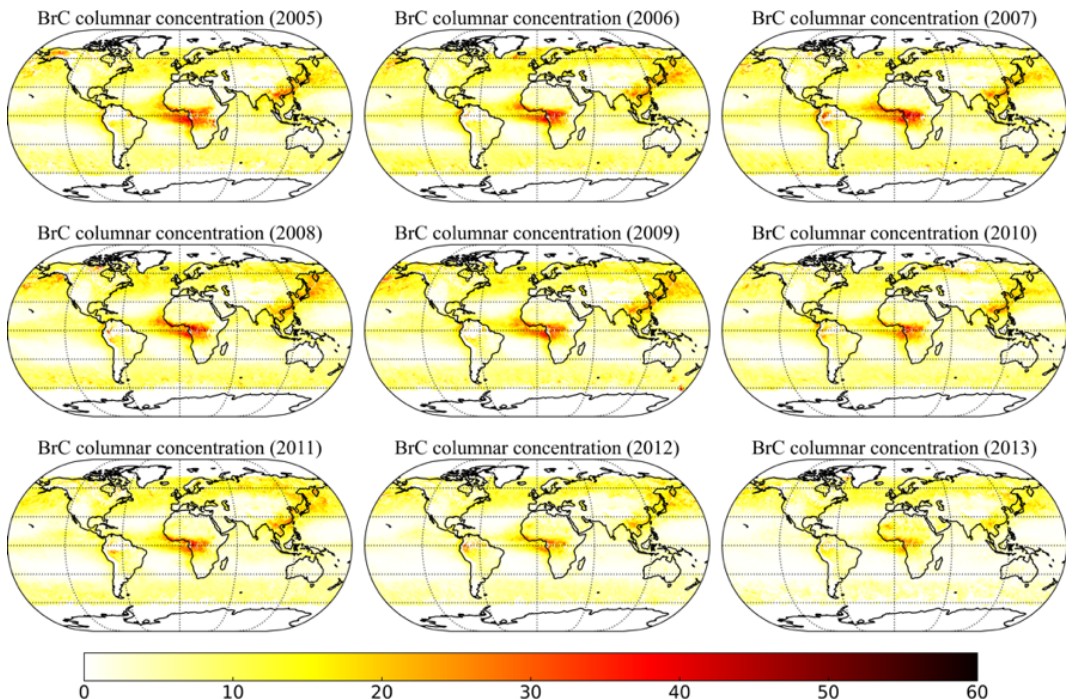

**Figure 7:** Yearly means of BrC columnar mass concentration (mg/m$^2$) derived by the GRASP/Component approach from POLDER-3 observations for the period from March 2005 to October 2013.


### 3.1.3 Coarse mode Absorbing Insoluble (CAI) aerosol component

Due to the absorption of hematite and goethite (free iron) contained in dust particles, mineral dust also indicates a spectral absorbing signature that stronger absorption at short wavelength and weaker absorption at long wavelength. In order to know the global distribution of free iron concentration for its

important biogeochemical and radiative impacts, coarse mode absorbing insoluble (CAI) component was identified as an important species in the GRASP/Component approach (Li et al., 2019). At the same time, because of a similar tendency in spectral absorption for BrC and CAI, the size-resolved component configuration that light-absorbing carbonaceous components (BC and BrC) are assumed to be absorbing component in the fine mode aerosol particles and CAI mainly representing iron oxides is assumed to be

absorbing component in the coarse mode aerosol particles was selected (Li et al., 2019). Fig. 8 shows the



satellite-derived seasonal CAI columnar mass concentration in the climatological products by the GRASP/Component for the period 2005 — 2013. As discussed by Li et al. (2019), CAI particles mainly represent iron oxides element contained in mineral dust. From Fig. 8, we can see that elevated CAI concentration are observed over the Sahara in western Africa, the Taklimakan in northwest China and over the Arabian Peninsula during the MAM and JJA seasons, which locations represent the global dust belt. Interesting to note that very low CAI aerosol concentrations are obtained over the Bodélé Depression, which indicates highly scattering dust particles (see the hot spot in Figs. 11 and 14 during the DJF season). This fact is consistent with the known dust composition nature of the Bodélé containing the dolomites mainly (Formenti et al., 2011). The maximum mass ratio of absorbing dust to scattering dust in our retrievals was found of about 5%, which is consistent with the known mass fraction of iron oxides varying from 3% to 5% in the desert dust measurements (Guieu et al., 2002; Zhang et al., 2003). It should be mentioned here however that our retrievals are constrained by the maximum value of iron oxides volume fraction of 3%. At the same time, this maximum value is practically never achieved indicating that the optical signal can be explained by iron oxides mass fraction within 5% (as the density is 4.8 g/cm$^3$ for iron oxides and 2.5 g/cm$^3$ for scattering dust component). The characteristics presented in the climatological CAI distribution (Fig. 8) are also consistent with the results of previous studies (Formenti et al., 2008; Krueger et al., 2004; Lázaro et al., 2008) reporting large proportions of iron oxides in dust originated from the Sahel belt between 0 and 20° N and Saudi Arabia, but low iron oxides content in the Chad Basin. The STD/Mean of seasonal CAI columnar mass concentration (Fig. 9) generally show high values in the desert dust regions that are mostly due to the significant changes in 2012 and 2013 shown by yearly means of CAI (Fig. 10).

It should be recalled that the CAI particles are parameterized in the GRASP/Component approach as particles having spectral absorption of iron oxides and this is the only the absorbing specie assumed in the coarse mode due to the limitations of measurements sensitivity. However, other light-absorbing species presenting in coarse-mode can produce similar signal and be erroneously interpreted as iron oxides. For instance, it can be noted that the CAI particles appear also in the regions and seasons associated with biomass burning, such as in South America during the SON season when strong biomass

burning events occur. The CAI in this case is therefore an indicator that carbonaceous particles might be
appearing in coarse mode by some mechanisms. Alternatively, some CAI particles appearing in South
America may be originating in dust transport from western Africa. This is in line with the previous studies
reporting the African dust transport over the Atlantic Ocean, Caribbean, Central America, and South
America (during the DJF and MAM seasons) (Griffin et al., 2002; Kalashnikova and Kahn, 2008;
Prospero and Lamb, 2003; Prospero and Mayol-Bracero, 2013). Analogously, some CAI particles
observed in northern and eastern China during the MAM season are transported from strong dust emission
sources in the Taklimakan desert. Based on the results of Li et al. (2019), the uncertainty in CAI fraction
associated with employed refractive index is within 50% excluding the case of very low CAI fraction,
below 0.005. Thus, the CAI concentrations are expected to have large uncertainties over ocean and in
high latitude near the polar region (such as the ocean around 60° S) where also the AOD is generally very
low. Also, cloud contaminations that are more probable in those high latitude areas can be misinterpreted
as apparent dust like aerosols.

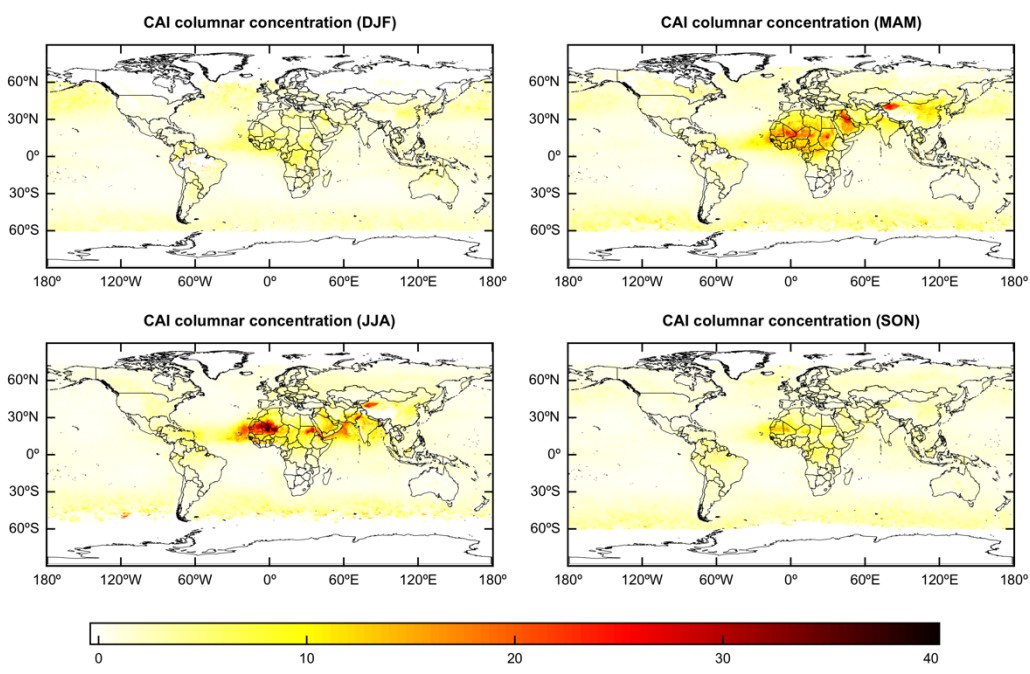

**Figure 8:** Spatial distribution of seasonal CAI (Coarse mode Absorbing Insoluble) columnar mass concentration (mg/m$^2$) derived by the GRASP/Component approach from POLDER-3 observations. DJF: December-January-February; MAM: March-April-May; JJA: June-July-August; SON: September-October-November.

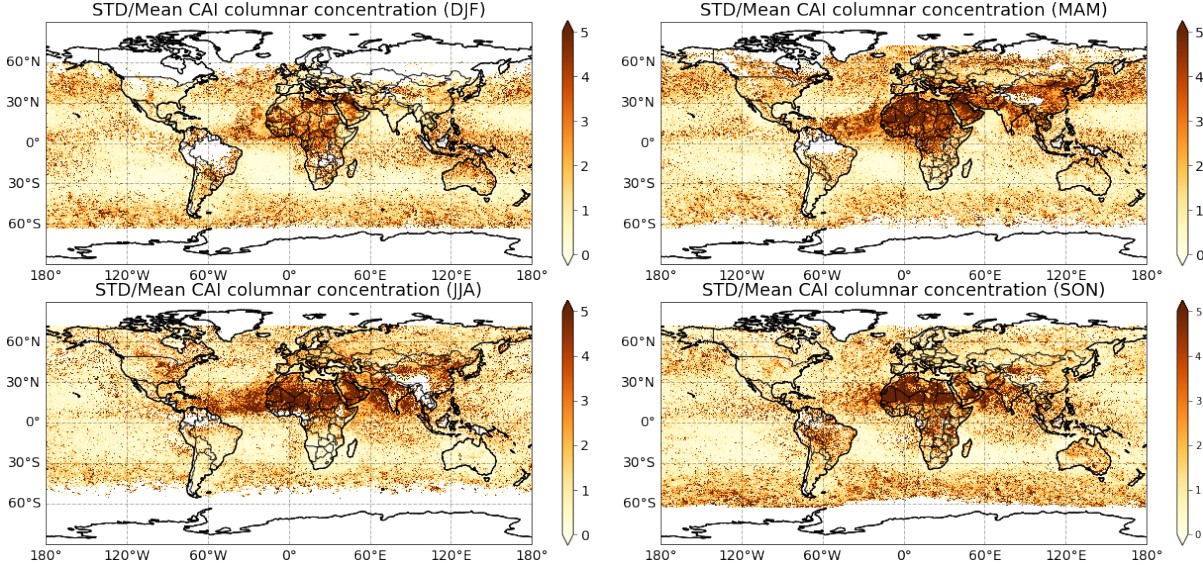

**Figure 9:** The standard deviation (STD)/Mean for seasonal CAI columnar mass concentration in Fig. 8 derived by the GRASP/Component approach from POLDER-3 observations. DJF: December-January-February; MAM: March-April-May; JJA: June-July-August; SON: September-October-November.



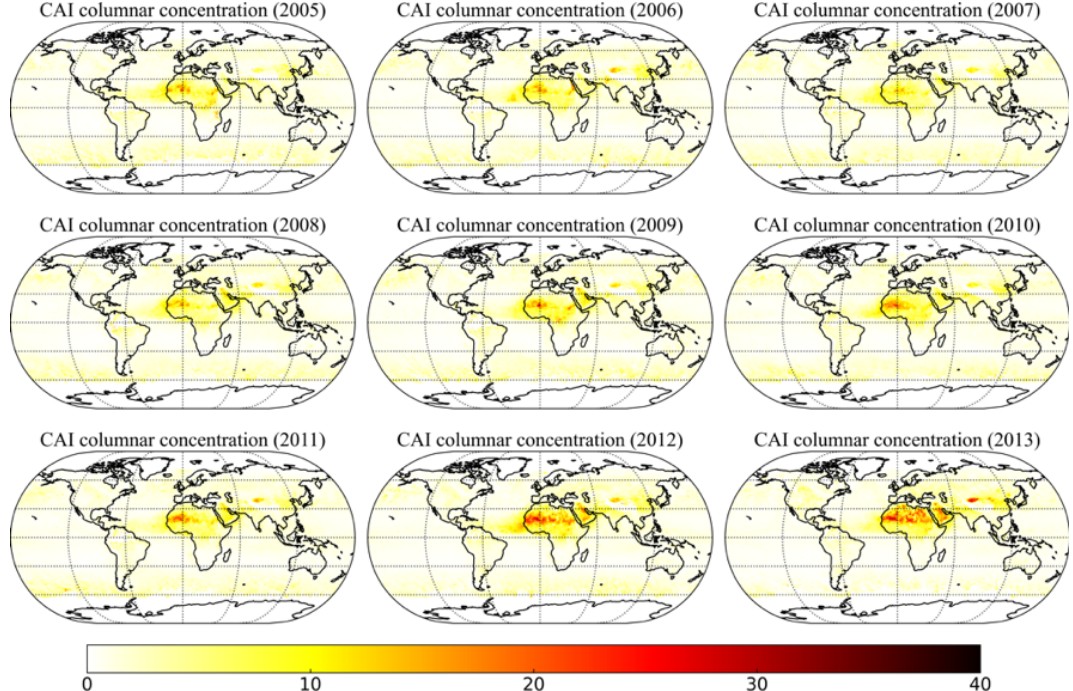


**Figure 10:** Yearly means of CAI columnar mass concentration (mg/m$^2$) in a resolution of derived by the GRASP/Component algorithm from approach from POLDER-3 observations for the period from March 2005 to October 2013.


### 3.2 GRASP/POLDER-3 scattering component products

As defined in Li et al. (2019), the weakly absorbing, mostly scattering aerosol components including coarse-mode non-absorbing insoluble component (CNAI), fine-mode non-absorbing insoluble component (FNAI), as well as Non-Absorbing Soluble (NAS) and Aerosol Water Content (AWC) in fine 495 and coarse mode are discussed below.

### 3.2.1 Coarse-mode Non-Absorbing Insoluble (CNAI)

The distribution of seasonal coarse-mode non-absorbing insoluble (CNAI) aerosol component and the associated STD/Mean are presented in Fig. 11 and 12, respectively. Yearly means of CNAI columnar



mass concentration for the period from March 2005 to October 2013 is presented in Fig. 13. Generally, the CNAI component represents the non-absorbing part of mineral dust particles. Fig. 11 and 13 reveal in details the global dust belt with noticeable maximum concentrations during the MAM and JJA seasons. The high STD/Mean values obtained over the main desert dust regions represent large interannual variability (Figs. 12 and 13). Specifically, the maximum CNAI concentration in the northern Africa shows a shift from lower to higher latitudes during the MAM and JJA seasons, which is associated with the change of the intertropical convergence zone. The most active dust emission source, Bodélé Depression, is seen well in the CNAI retrievals (shown in Fig. 11) with the peaks during the DJF and MAM seasons, consistent with studies of Todd et al. (2007) and Washington et al. (2003). High coarse-mode non-absorbing dust concentrations are observed over the Arabian Peninsula and Taklamakan in China during the MAM and JJA seasons while significant decrease appears during the SON and DJF seasons. In addition, a key dust emission source in southwest Asia, named Sistan region (Goudie, 2014; Léon and Legrand, 2003; Middleton, 1986), can be seen in Fig. 11 during the JJA season.

The well-known locations of source regions of mineral dust over the globe are reported by numerous studies: the Sahara region in the northern Africa (e.g., Choobari et al., 2014; Prospero et al., 2002); the Middle East and Arabian Peninsula (Ginoux et al., 2012; Prospero et al., 2002); the Sistan Basin (representing the region of Iran, Pakistan, Afghanistan); the desert regions in southwestern Asia (Alizadeh-Choobari et al., 2014; Kaskaoutis et al., 2015); the Taklamakan in China and Gobi (northern China–southern Mongolia) deserts (Ginoux et al., 2012); as well as several desert regions in central Asia (Elguindi et al., 2016) and in North America (Ginoux et al., 2012). All of them represent the Northern Hemisphere as location of the most important natural dust emission sources. Till present, the studies of these dust sources were conducted based on satellite remote sensing mostly providing the total AOD, coarse mode AOD or AOD related, under extra assumptions, to desert Dust Optical Depth (DOD) (Logothetis et al., 2021). Those studies were conducted based on derived dust indices, AOD, aerosol profiles and on thermal infrared signal, i.e. from products of Total Ozone Mapping Spectrometer (TOMS, Ginoux et al., 2012; Prospero et al., 2002), MODIS (Gui et al., 2021c; Remer et al., 2008; Schepanski et al., 2012; Song et al., 2021; Voss and Evan, 2020; Yu et al., 2019), Infrared Atmospheric Sounding

Interferometer (IASI, Clarisse et al., 2019; Yu et al., 2019), MISR (Gui et al., 2021a; Kahn and Gaitley, 2015; Yu et al., 2019), Cloud–Aerosol Lidar with Orthogonal Polarization (CALIOP, Gui et al., 2021b, 2021c; Shikwambana and Sivakumar, 2018; Song et al., 2021). In this regard, the GRASP/Component

product from POLDER-3 privies certainly more direct quantitative evaluation of the mineral dust composition and its iron oxide content enabling studies of the trends and variability. For example, as presented in Fig. 13, over the Arabian Peninsula, the lower CNAI concentrations are obtained for 2005 – 2007 and higher values CNAI concentrations are obtained for 2008 – 2013 periods, which agrees well with the results of Notaro et al. (2015) indicating an inactive and active dust periods (1998 – 2005 and

2007 – 2013, respectively) over the Arabian Peninsula related to the sustained drought near the Fertile Crescent region. MODIS DOD products indicated that the dust loadings have an increase over the Sahara region during the period 2003 – 2018 (Voss and Evan, 2020), which can be seen now quantitatively in Fig. 13.

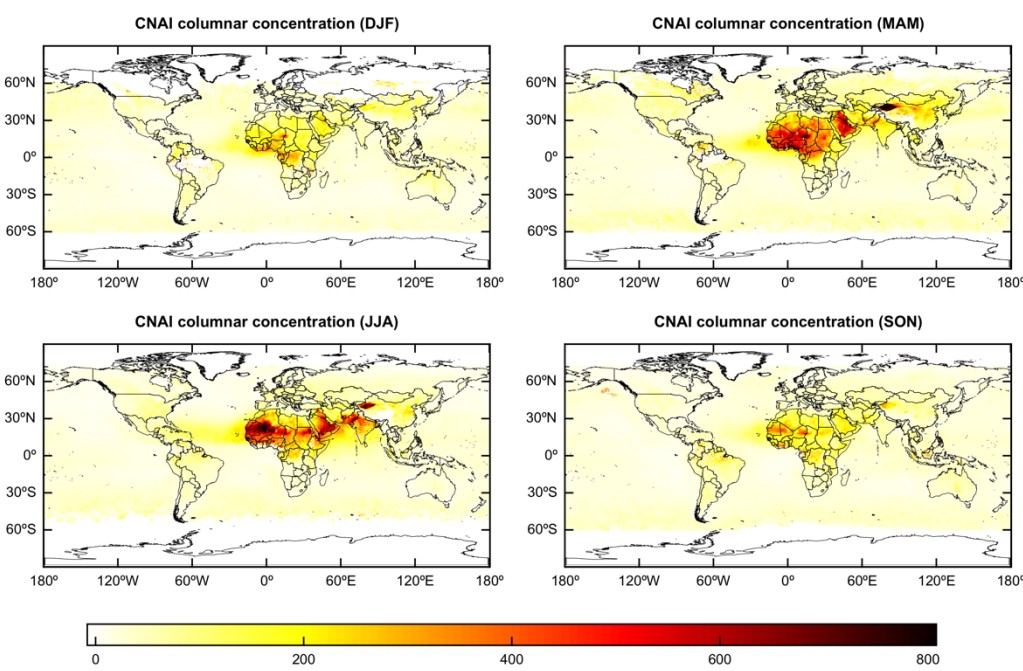

**Figure 11:** Spatial distribution of seasonal CNAI (Coarse-mode Non-Absorbing Insoluble component) columnar mass concentration (mg/m$^2$) derived by the GRASP/Component from approach from POLDER-



3 observations. DJF: December-January-February; MAM: March-April-May; JJA: June-July-August;
SON: September-October-November.

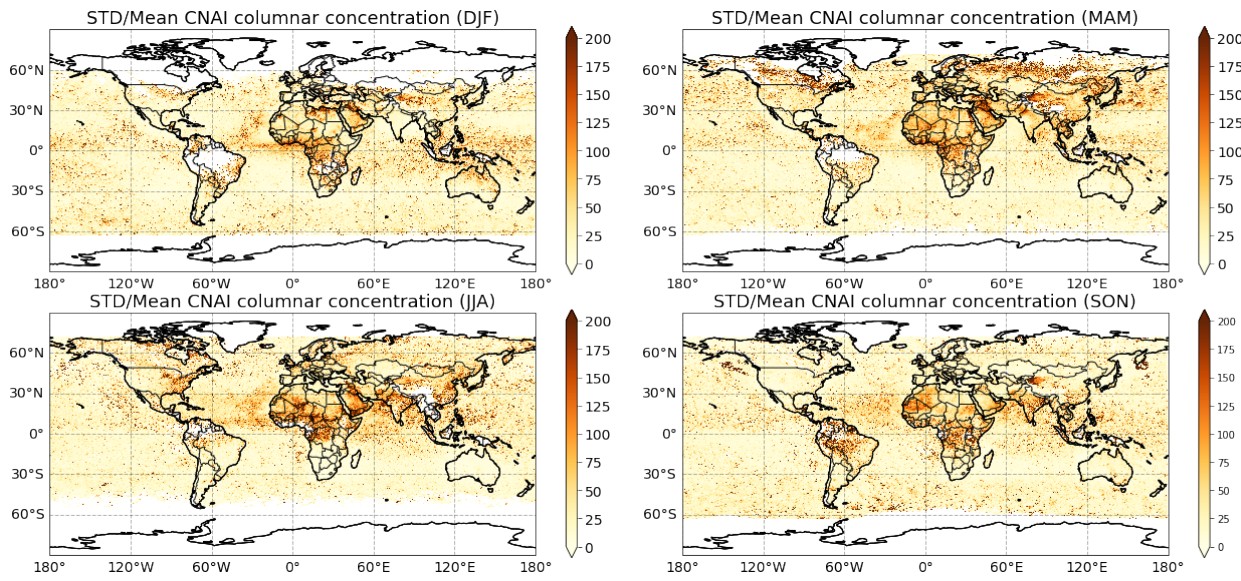


**Figure 12:** The standard deviation (STD)/Mean for seasonal CNAI columnar mass concentration in Fig. 11 derived by the GRASP/Component from approach from POLDER-3 observations. DJF: December-January-February; MAM: March-April-May; JJA: June-July-August; SON: September-October-November.


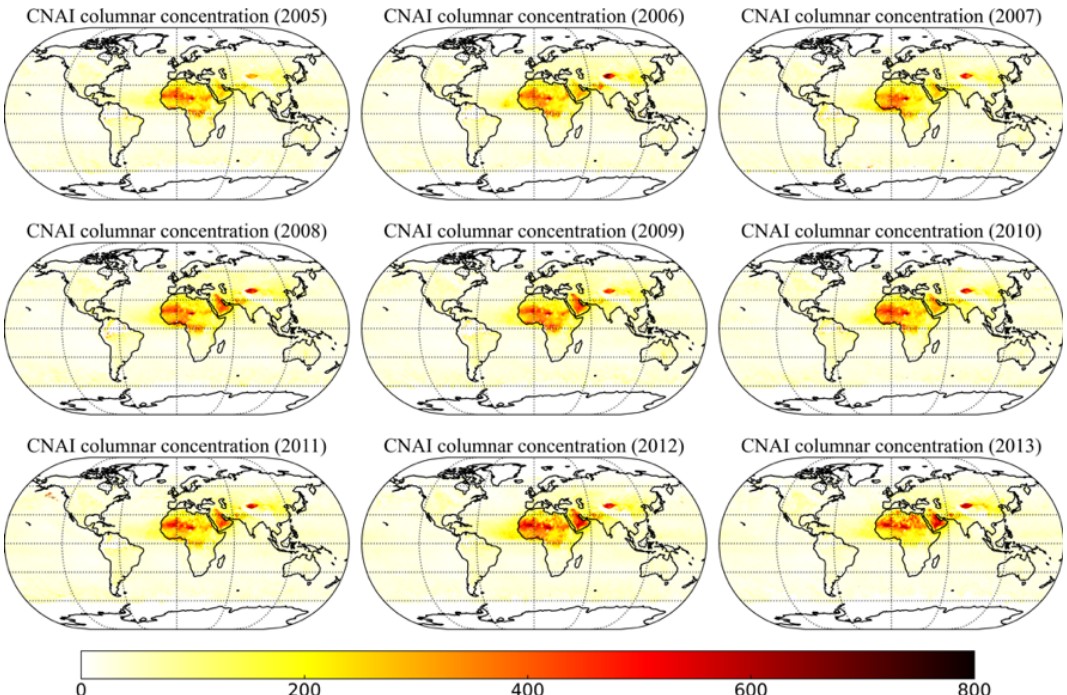

**Figure 13:** Yearly means of CNAI columnar mass concentration (mg/m$^2$) in a resolution of 0.1° × 0.1° derived by the GRASP/Component from approach from POLDER-3 observations for the period from March 2005 to October 2013.

### 3.2.2 Fine-mode Non-Absorbing Insoluble (FNAI)

Figure 14 shows the pattern of seasonal fine-mode non-absorbing insoluble component (FNAI) columnar mass concentration in the derived climatology, as well as the STD divided by the mean of FNAI concentration in Fig. 15. As aforementioned and discussed by Li et al. (2019), both scattering organic carbon and non-absorbing dust particles in fine-mode aerosol are labelled as FNAI in current GRASP/Component. Thus, significant similarities and differences are obviously seen between FNAI (Fig. 14) and CNAI (Fig. 8) spatio-temporal distributions. Specifically, similar CNAI and CAI, the spatio-temporal characteristics of global dust belt appears in the FNAI. That is, the concentrations appear over northern Africa, Bodélé Depression, Arabian Peninsula, and the Taklimakan Desert. However, the differences are manifested in well-known for the biomass burning regions and seasons, i.e. in southern

Africa and South America during the JJA and SON seasons, in the Indo-China Peninsula during the MAM
season, as well as fossil fuel burning associated with strong anthropogenic activities in China and India.
The retrieved FNAI particles, mainly representing the fine-mode scattering organic carbon, present good
spatio-temporal consistency with the results of previous studies on the variability of anthropogenic and
biomass burning emissions (Duncan et al., 2003; Zhang et al., 2008b, 2012). Large STD/Mean values are
obtained over southern Africa and South America which can be attributed to the interannual variations of
biomass burning events. For example, the yearly means of FNAI columnar mass concentration shown in
Fig. 16 presents extremely high FNAI concentration over South America in 2005, 2007, and 2010, which
could be caused by the areas and frequency of El Niño-induced droughts and fires in this region. The
uncertainty of the FNAI retrievals depends on the aerosol loading and is generally be less than 100%.

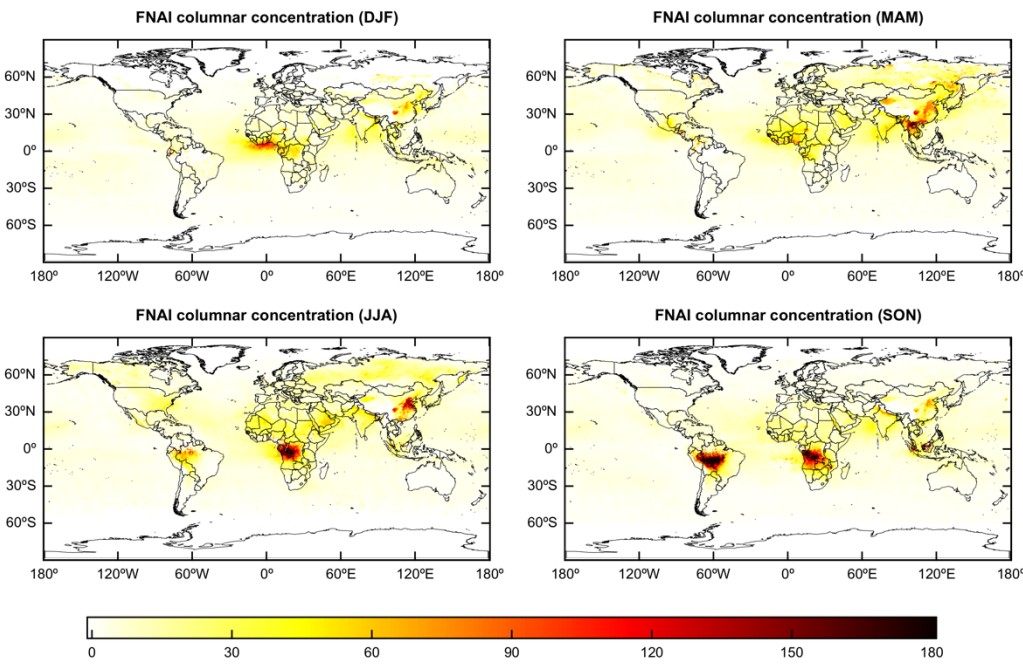

**Figure 14:** Spatial distribution of 0.1° × 0.1° seasonal FNAI (Fine mode Non-Absorbing Insoluble
component) columnar mass concentration (mg/m$^2$) derived by the GRASP/Component from approach
from POLDER-3 observations. DJF: December-January-February; MAM: March-April-May; JJA: June-
July-August; SON: September-October-November.

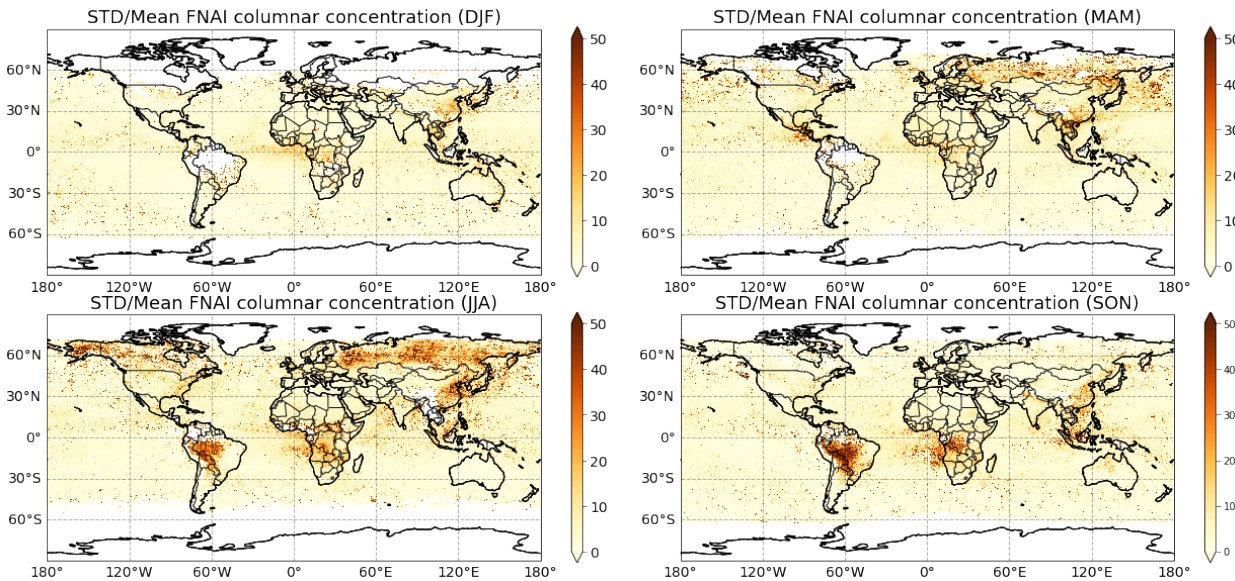

**Figure 15:** The standard deviation (STD)/Mean for seasonal FNAI columnar mass concentration in Fig. 14 derived by the GRASP/Component from approach from POLDER-3 observations. DJF: December-January-February; MAM: March-April-May; JJA: June-July-August; SON: September-October-November.

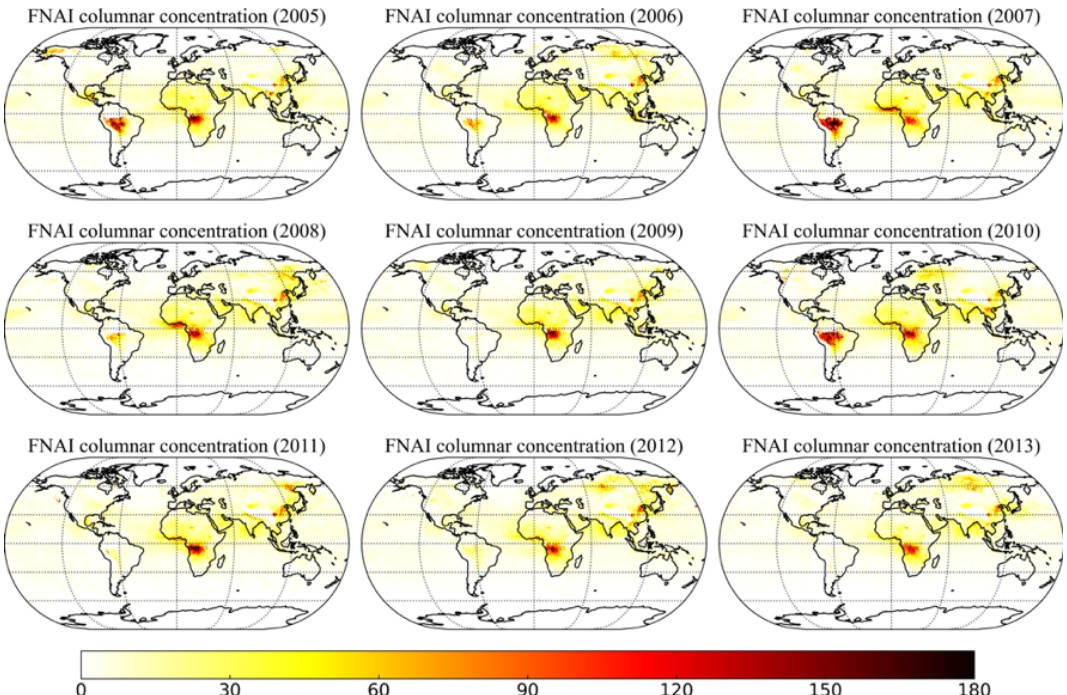

**Figure 16:** Yearly means of FNAI columnar mass concentration (mg/m²) in a resolution of 0.1° × 0.1° derived by the GRASP/Component from approach from POLDER-3 observations for the period from March 2005 to October 2013.


### 3.2.3 Non-Absorbing Soluble (NAS) and Aerosol Water Content (AWC)

As aforementioned and discussed in Li et al. (2019), the components contained in the host can represent all of ammonium nitrate, sulfate, sea salt and aerosol water content in fine- and coarse-mode aerosol
fraction. However, the NAS also can be aged hydroscopic components such as water-soluble organic carbon generated from biomass burning or anthropogenic activities, and water-soluble mineral ions provided by dust particles. The NAS and AWC components thus can be quite complex for interpretation, however, overall the obtained climatology presents a reasonable logic. For example, fine-mode non-absorbing soluble component (FNAS, in Fig. 17) is high in China and India, especially during the DJF
and SON season that is in line with strong anthropogenic emissions (e.g., industrial and heating activity)



in megacities where the population is dense. The non-absorbing water-soluble organic carbon, that can be emitted during biomass burning events, can be also detected and interpreted as FNAS, e.g., as it occurs in southern Africa. The GRASP/Component retrievals also indicate many NAS particles present in the Mediterranean region, which is in line with the previously demonstrated aerosol mixture constituted of

anthropogenic aerosols, sea salt particles, and mineral dust coated with biogenic sulfate (Ganor et al., 2000; Lelieveld et al., 2002; Levin, 2005; Levin et al., 1996). The spatio-temporal distributions of aerosol water content appear to be similar to the non-absorbing soluble component, which is quite logical because these two components are naturally related and also form the particle host in the employed Maxwell-Garnet effective medium approximation mixing rule in the algorithm. The fine-mode aerosol water

content (Fig. 18) is also naturally associated with hygroscopic anthropogenic aerosol particles and dominates in regions such as Asia. The aerosol water presence over ocean during the dust transport and near southern Africa during the biomass burning is also rather logical because of higher atmospheric relative humidity.

Moreover, possible the hygroscopicity of aged dust was reported in several studies (Sullivan et al., 2009;

Tang et al., 2016) and the signal can also appear in observations (Derimian et al., 2017; Falkovich et al., 2004). Besides some regions with high coarse-mode non-absorbing insoluble (dust) concentration over ocean (near western Africa and Arabian Peninsula), the values of NAS and AWC in coarse mode are rather low everywhere. We note that a relatively low fraction but with very large volume concentration could produce relatively important concentration. In addition, in situations when the real part of aerosol

refractive index is very low, the algorithm may try to introduce a water fraction and thus produce an artefact. This is rather the case when a water fraction is retrieved over the Bodélé Depression during the DJF season. Indeed, an important proportion of fossil diatom in the sediments of the Bodélé Depression (Formenti et al., 2008) can produce dust with unusually low real part of refractive index. To reproduce the optical signal of this dust, a small fraction of water can be required (element with the lowest real

refractive index). Since the dust concentrations are extremely high in this source, even small fraction presents relatively high water concentrations, which is an artefact. This example illustrates that interpretations of the retrieved component fractions are not always straightforward and depend on limits

in the measurement sensitivity. Nevertheless, the measurements reflecting physical properties and a closer analysis can usually explain mismatches with expectations.

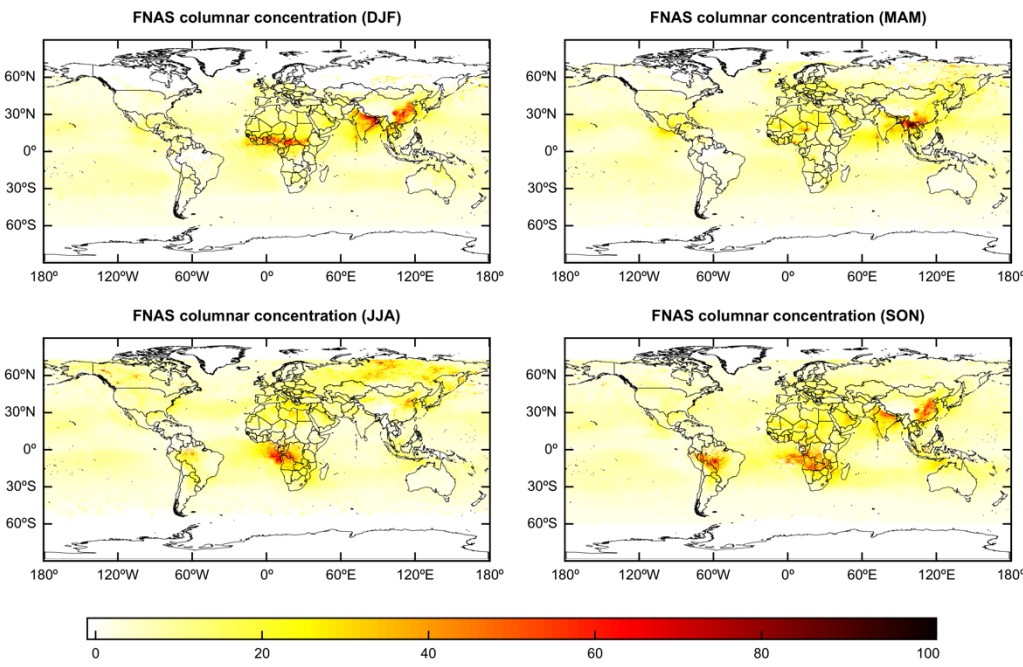


**Figure 17:** Spatial distribution of 0.1° × 0.1° seasonal FNAS (Fine mode Non-Absorbing Soluble component) columnar mass concentration (mg/m$^2$) derived by the GRASP/Component from approach from POLDER-3 observations. DJF: December-January-February; MAM: March-April-May; JJA: June-July-August; SON: September-October-November.


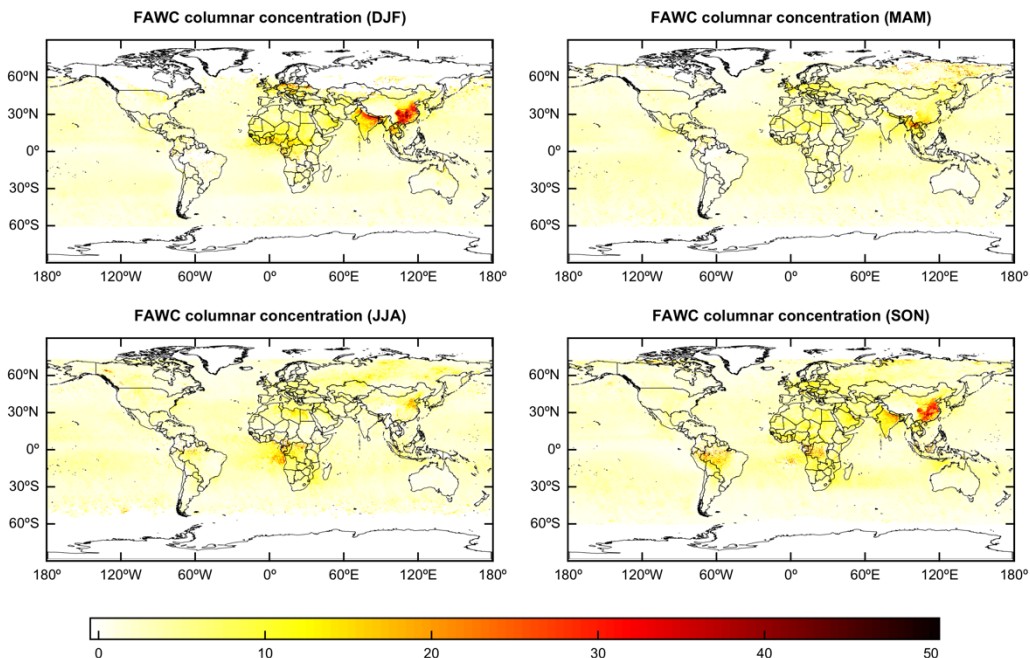

Figure 18: Spatial distribution of 0.1° × 0.1° seasonal FAWC (Fine mode Aerosol Water Content) columnar mass concentration (mg/m$^2$) derived by the GRASP/Component from approach from POLDER-3 observations. DJF: December-January-February; MAM: March-April-May; JJA: June-July-August; SON: September-October-November.

## 4 Comparisons of BC and dust concentration derived by GRASP/Component with MERRA-2

Although applications of GRASP/Component to different instruments (sun photometer, POLDER-3, DPC/GF-5) have presented good consistency with the available AERONET aerosol optical products such as AOD, AAOD, AE and also agree with the physical expectations about aerosol composition in different locations, an addition reference data are still desirable for the validation. Because the in situ measurements data are extremely scarce and difficult for global validation, we employ an inter-comparison of BC and dust columnar concentration between GRASP/Component climatological dataset and the corresponding MERRA-2 products. Indeed, this is not a pure validation experiment, but can still provide a better



indicator of a global consistency. The CNAI component here we will refer to the dust component in MERRA-2. The GRASP/Component dust and also BC products will be inter-compared. To this end, the GRASP/Component products are re-gridded into the same resolution of 0.5° × 0.625° as of MERRA-2. Fig. 19 presents the differences of monthly BC columnar concentration between GRASP/Component and MERRA-2 for the period March 2005 – October 2013. One can note that both GRASP/Component BC

and MERRA-2 BC represent quite similar spatio-temporal pattern and generally show reasonable consistency between them over ocean, Europe, United States, and mostly over South America and Africa, except however some noticeable differences during strong biomass burning season and strong anthropogenic emissions in China and India. In further discussion, we analyze the possible source of these discrepancies.


First, MERRA-2 includes the estimations of several externally mixed aerosol components (e.g., BC, OC, dust, etc.) as defined in the Goddard Chemistry, Aerosol, Radiation, and Transport model (GOCART) (Chin et al., 2002). The emissions for carbonaceous aerosol species in GOCART include fossil fuel combustion, biomass burning, biofuel consumption, and biogenic sources. Both natural and

anthropogenic sources make contributions to BC emissions in MERRA-2. Various inventories during the different time periods were considered to produce biomass burning emissions of carbonaceous aerosols in MERRA-2. Specifically, the Quick Fire Emissions Dataset (QFED) version 2.4-r6 (Darmenov and da Silva, 2015) has been employed for daily biomass burning BC emission since 2010. The monthly biomass burning emissions in Global Fire Emission Dataset (GFED) version 3.1 (Randerson et al., 2006; Van Der

Werf et al., 2006) were used for the period 1997 – 2009. Biome-dependent factors are used to make corrections between GFED v3.1 and QFED v2.4-r6 during the period 2003 – 2011 when they were available. On the other hand, it is noted that more and more strict controls on air pollution in China have taken with the rapid economic growth (Zhang et al., 2019). The inventory of Peking university (PKU) indicated anthropogenic aerosol emissions present decrease in the key regions of China since 2006 with

a significant trend of -1.4% per year for anthropogenic BC emissions during the period 2006 – 2014 (Wang et al., 2021). However, a continuous increase of anthropogenic aerosol emissions in China from 2006 to 2014 is shown in the Community Emissions Data System (CEDS) inventory, thus the global





climate models cannot capture the observed downward trend well (Wang et al., 2021). The comparisons of biomass burning emission datasets including GFED3.1 (Global Fire Emissions Database version 3.1),

GFED4s (GFED version 4 with small fires), FINN1.5 (FIre INventory from NCAR version 1.5), GFAS1.2 (Global Fire Assimilation System version 1.2), FEER1.0 (Fire Energetics and Emissions Research version 1.0), and QFED2.4 (Quick Fire Emissions Dataset version 2.4) for 2008 globally have indicated that a factor of 3.4 can be obtained on an annual average for the differences of BC emissions (Pan et al., 2020). At the same time, the knowledge on the complexity and variability of different fire

features are still required to be improved (Hyer et al., 2011). Continuous measurements are required to contribute toward advancing our understanding of the fire-generated BC globally, at least in the major biomass burning regions. The emission factors for different ecosystem types and burning stages are needed to be estimated accurately for the improvement of emission globally (Pan et al., 2020).

Second, a higher value of BC refractive index ($1.95 - 0.79i$) employed in the GRASP/Component for the

BC retrievals (Li et al., 2019) than BC refractive index ($1.75 - 0.45i$) used in the GOCART model (Chin et al., 2002), which affects the GRASP BC mass retrievals, also makes contributions to the BC differences. Finally, we also should note that a new light-absorbing carbonaceous component, named BrC, is included in the GRASP/Component products so that the spectral aerosol absorption signature in the fine-mode fraction can be interpreted as BrC contribution in the satellite inversion. Whereas at present

MERRA-2 do not provide BrC product. Therefore, for instance, the noticeable BrC concentrations over ocean near the southern Africa for the strong biomass burning season (in Fig. 5) can indeed explain the BC underestimation by GRASP/Component relative to MERRA-2.

The monthly dust columnar concentration differences between GRASP/Component and MERRA-2 for the period March 2005 – October 2013 are presented in Fig. 20. We can see that GRASP/Component dust

concentration is less than one of MERRA-2 over the northern Africa, Arabian Peninsula, Middle Asia, and northwest China. We would like to recall that the wind speed-dependent emissions with the potential dust source locations described in Ginoux et al. (2001) are used for dust missions in the MERRA-2/GOCART model and that the depositions (such as dry deposition, wet removal globally) of aerosol are not considered in the current GOCART estimation for MERRA-2 (Randles et al., 2017). These factors

can cause the overestimation of dust in MERRA-2. At the same time, the GRASP/Component dust

concentration used for the comparison represent the CNAI (coarse-mode non-absorbing insoluble component) concentration that excludes the fine-mode dust and absorbing that leads to a systematic underestimation of the dust component as it is defined in MERRA-2. These differences can merit a closer future analysis as function of dust source region, composition and transport distance.


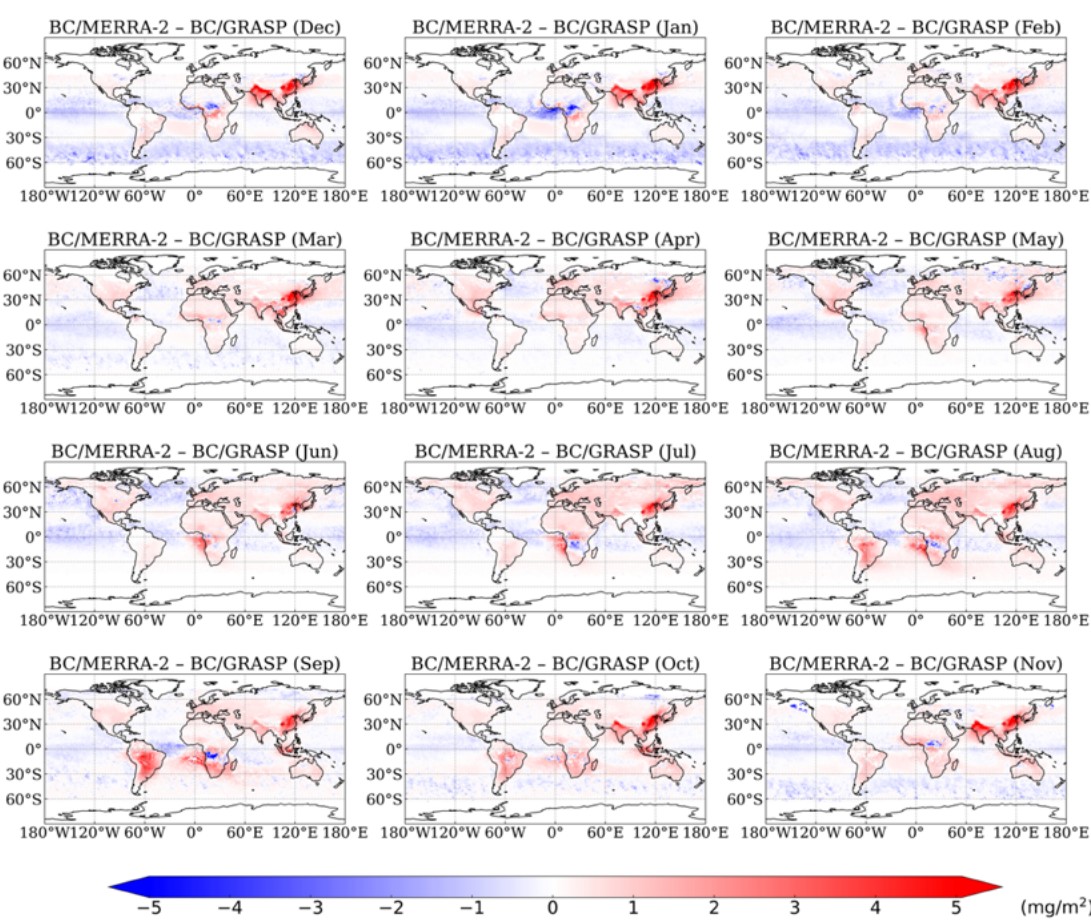

**Figure 19:** Differences of monthly BC columnar concentration between GRASP/Component and MERRA-2 in the resolution of 0.5° × 0.625° for the period March 2005 – October 2013.

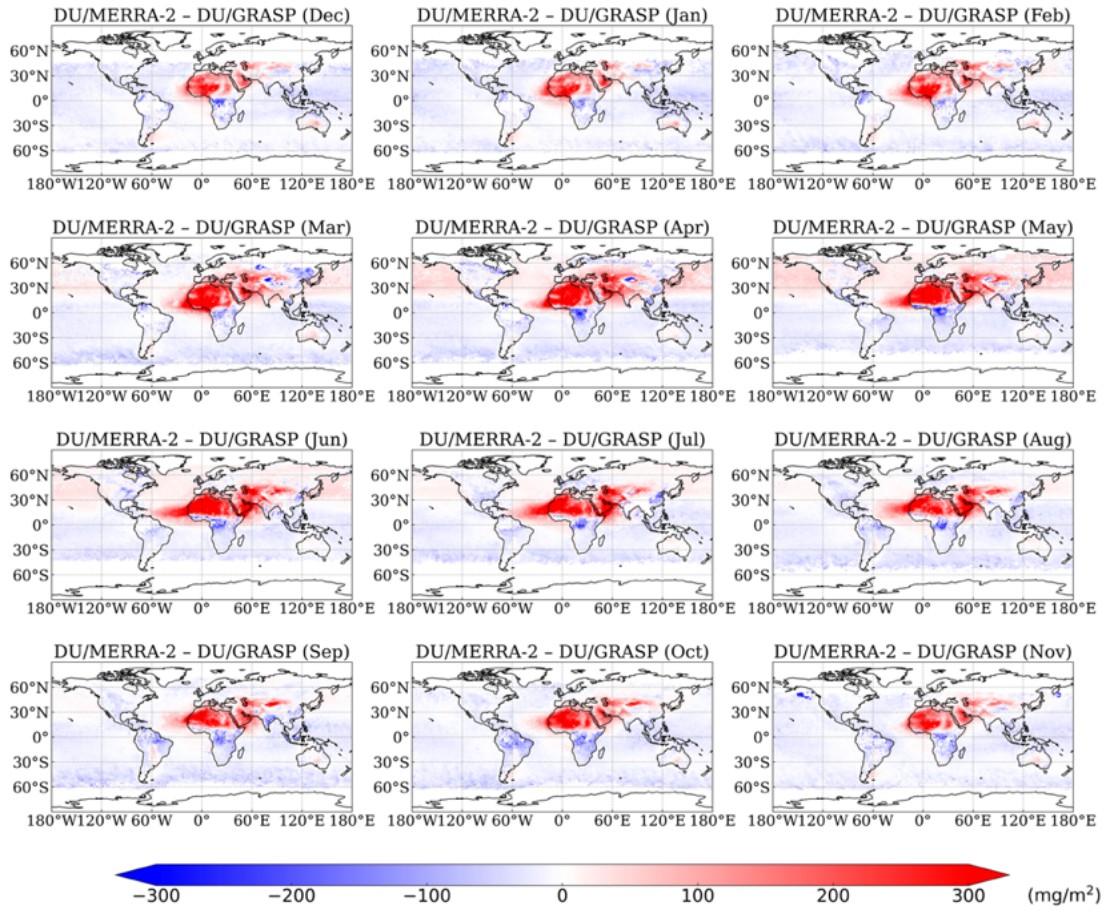

**Figure 20:** Differences of monthly dust columnar concentration between GRASP/Component and MERRA-2 in the resolution of 0.5° × 0.625° for the period March 2005 – October 2013.

## 5 Data availability

The GRASP/Component products derived from POLDER-3 are publicly available on the GRASP algorithm website (https://www.grasp-open.com/products, last access: 15 March 2022). The monthly MERRA-2 BC and dust columnar concentration products are available at NASA website (https://daac.gsfc.nasa.gov). The dataset used in the current study is registered under https://doi.org/10.5281/zenodo.6395384 (Li et al., 2022).

## 6 Conclusions

The current paper presents a climatological analysis of the aerosol product recently generated from POLDER-3 by newly developed GRASP/Component approach. This approach employs an essentially
different, compared to previous algorithms, methodology and attempts to derive not only optical properties of aerosol and also some information about aerosol composition. The GRASP/Component algorithm provides global satellite observations based concentrations of absorbing components such as BC, BrC, CAI representing iron oxides contained in dust, and of scattering components such as fine mode organic carbon/scattering dust and coarse mode scattering dust, which is a scarce but imperative
information required for tuning the chemical transport models. This GRASP generated satellite-derived aerosol component product is expected to be valuable for qualitative and quantitative understanding of aerosol components distribution on global and regional scales. More importantly, the data assimilation of the derived fractions of different aerosol components into the chemical transport models can provide an important constraint for improving the aerosol modeling simulations and reduce some regional
inconsistency among climate models. This paper show and disuses the climatological patterns derived from POLDER-3 using GRASP/Component approach. It is shown that this global satellite derived aerosol chemical composition climatology clearly reveals the distribution of global dust belt, biomass burning events, and anthropogenic activity emissions that is in good agreement with known logical expectations and previous studies. The inter-comparisons of BC and dust columnar concentrations provided by
GRASP/Component and MERRA-2 for the period March 2005 – October 2013 show good agreement in global spatio-temporal patterns, although some explainable differences appear for strong biomass burning events and in regions strongly affected by mineral dust. For example, the BC emissions can affect significantly the MERRA-2 BC concentrations during the strong biomass burning periods.

In addition, all GRASP/Component generated optical properties agree very closely with the best previously obtained POLDER-3 aerosol products not only over AERONET stations and also globally both over land and over ocean. Specifically, the GRASP/Component approach is demonstrated to provide the aerosol optical characteristics of comparable or sometimes even better accuracy (as validated by AERONET data) as all other earlier GRASP/Optimized, High Precision and Models products. For





example, the validation study by Zhang et al. (2021b) using global AEROENT data shown that total spectral AOD quite closely agrees with GRASP/Models retrieval that has highest correlation and lowest bias (Chen et al., 2020). At the same time, most of the GRASP/Component detailed characteristics (spectral AODF, AODC, AAOD and SSA) provided by GRASP/Component also agree well with the aerosol optical products of previous GRASP approaches (Optimized, High Precision, and Models) in Chen et al. (2020).

Although a rigorous global validation of aerosol component remains challenging at the current stage, its estimation is the next frontier for the aerosol inversion from satellite observation because is required for continuous evaluation of aerosol life cycle by chemical transport models. Also, it should be pointed out that in situ aerosol component data are scarce unlike satellite AOD products on global scale. The presented here satellite aerosol component product is comparable in coverage with conventional AOD product, whole retrieval require more complex aerosol parameterization and advanced remote sensing observations such as the multi-angular polarimetric observations. At the same time, many space-borne instruments with the multi-angle polarization measurement capabilities are planned to be deployed and available in the next few years (Dubovik et al., 2019), such as 3MI (Multi-View Multi-Channel Multi-Polarization Imaging), DPC-02 (Directional Polarimetric Camera) etc. Finally, the presented work illustrates that with the synergy of different remote sensing instruments and additional algorithm refinements allow the extensive retrievals of aerosol microphysical and optical properties together with aerosol composition information, as derived by GRASP/Component algorithm, can be extensively generated in the near future.

**Author contributions.** The GRASP/Component approach was developed by the GRASP team (OD, LL, YD, GS, DF, PL, TL, AL, CC, FD, YK, BT). LL, YD, HC and OD carried out this study and analysis. CC, XZ, CM, KG, YZ and YL contributed to the post-processed retrievals and the figures. The component products were widely discussed with some modeler, who are co-authors of this paper. LL, HC and OD wrote the manuscript with contributions from all authors.





**Competing interests.** The authors declare that they have no conflict of interest.

**Acknowledgements.** The authors would like to acknowledge the use of POLDER-3 Level-1 data originally provided by CNES. The authors are also grateful to the MERRA-2 product team.

**Financial support.** This research has been supported by the National Science Fund for Distinguished Young Scholars (41825011), the National Natural Science Foundation of China (41905117 & 42030608), the National Key Research and Development Program (2019YFC0214603). The component algorithm was developed as part of the Labex CaPPA project, which is funded by the Agence Nationale de la Recherche (grant no. ANR-II-LABX-0005-01).

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
