# Peer review of "Climatology of aerosol components concentration derived by GRASP algorithm from multi-angular polarimetric POLDER-3 observations"

_Earth System Science Data, 2022_

## Author Response (AR1)

**Reviewer #1**:

This study presents the climatology of aerosol mass burden for different components based on a newly developed algorithm GRASP/Component with the input of POLDER-3 multi-channal radiance observations. Global observations of aerosol component concentration are crucial for both the estimation of aerosol effective radiative forcing from satellite observations and also the evaluation of the performance of global aerosol models, but retrieving those information from satellite has been challenging. Most of previous studies just used the column aerosol optical properties (e.g., AOD/AODf/AI) to estimate aerosol forcings, it seems somewhat rough but is the only feasible option. The dataset presented here is thus very valuable and useful for our scientific community. This manuscript is well organized, the analysis methods are technically sound. I personally think this manuscript is suitable for publication after a minor revision.

**Response:** Thank you very much for the time and efforts you have put into reviewing the manuscript. We are very grateful for your positive evaluations and helpful comments on our work, which have enabled us to improve the manuscript. Here are our point-by-point responses to the comments:

**Detail comments:**

1. Figures showing annual mean concentrations of different aerosol components (2005 - 2013) are quite interesting, but not much mentioned in the main text. Many recent studied revealed the aerosol effective forcing changed during last two decades due to the change in anthropogenic emissions (https://doi.org/10.5194/acp-2022-295; https://doi.org/10.5194/essd-12-1649-2020). I feel that it could be quite interesting to plot the long-term trends of different aerosol components over main industrial regions (West Europe, East U.S., China and India) and discuss the potential implication or linkage to the observed radiative forcing trend.

**Response:** We appreciated the reviewer's comment and suggestion very much. As the referee mentioned, the linear trends of different aerosol components and also the potential implication or link to the observed radiative forcing trend are highly

interesting and merit a more focused effort. We also agree that a regionally zoomed analysis of the linear trends of components, as well as their potential implications, should be in scope of future studies. At present, due to very large volume of produced data in this publication and limited resources, we have added only a global overview of linear trends of components. We also added a discussion on potential implications, including two citation (see below), in order to highlight the scientific significances of the presented in our study component retrievals.

We have added the text in the abstract of the revised version (lines 43-45): "The extensive satellite-based aerosol component dataset is expected to be useful for improving global aerosol emissions and component-resolved radiative forcing estimations."

Also, lines 807-821 and 865-870 in the revised version: "More importantly, the data assimilation of this extensive satellite-based aerosol component dataset can importantly contribute to improving global aerosol emissions estimation and further improvement of accuracy of the estimated aerosol radiative forcing in general and per aerosol component in particular. For instance, the AOD and the AAOD products derived from POLDER-3 observations have been already used to constrain GEOS-Chem inverse modeling for the improvement of global black carbon, organic carbon and desert dust aerosol emissions (Chen et al., 2018, 2019). Using the presented in this study additional satellite-based aerosol component products, the further improvement of global aerosol emissions estimation is thus expected. The presented efforts are also in line with the studies suggesting that the employment of satellite-constrained anthropogenic and natural aerosol emissions by the climate models is required to improve the accuracy of aerosol radiative forcing estimations (e.g., Bellouin et al., 2020; Quaas et al., 2022). That is, the linear trends in column concentration of the main aerosol components, such as BC, BrC, CAI and CNAI as shown in Fig. S1 in the supplement, can provide a better global scale satellite-measured constraints on the properties of anthropogenic (BC and BrC) and natural (CAI and CNAI) aerosols and will contribute to improving the accuracy of anthropogenic aerosol radiative forcing estimations."

[Figure]

Fig S1. Linear trends in column concentration of BC, BrC, CAI and CNAI components with the criteria of AOD (440 nm) > 0.2 and for BC (> 1 mg/m$^2$), BrC (> 10 mg/m$^2$), CAI (> 2 mg/m$^2$), and CANI (> 50 mg/m$^2$).

Reference:

Bellouin, N., Davies, W., Shine, K. P., Quaas, J., Mülmenstädt, J., Forster, P. M., Smith, C., Lee, L., Regayre, L., Brasseur, G., Sudarchikova, N., Bouarar, I., Boucher, O., and Myhre, G.: Radiative forcing of climate change from the Copernicus reanalysis of atmospheric composition, Earth Syst. Sci. Data, 12, 1649–1677, doi: 10.5194/essd-12-1649-2020, 2020.

Quaas, J., Jia, H., Smith, C., Albright, A. L., Aas, W., Bellouin, N., Boucher, O., Doutriaux-Boucher, M., Forster, P. M., Grosvenor, D., Jenkins, S., Klimont, Z., Loeb, N. G., Ma, X., Naik, V., Paulot, F., Stier, P., Wild, M., Myhre, G. and Schulz, M.: Robust evidence for reversal in the aerosol effective climate forcing trend, Atmos. Chem. Phys. Discuss., 1–25, doi: 10.5194/acp-2022-295, 2022.

2. Line 281: How can the authors conclude 'the GRASP/Component provided the overall most consistent both total and detailed aerosol properties' based on the findings by Zhang et al. (2021)? I don't understand the causal link. Could the authors develop a bit on this?

**Response:** We appreciated the reviewer's suggestion very much. We added more descriptions and explanations in the revision (Lines 311-317): "Indeed, Zhang et al. (2021) validated GRASP/Component optical properties against AERONET data and concluded that generated total AOD values have minimal bias both over land (-0.02 for 550 nm) and ocean (0.01 for 550 nm), similar to total AOD provided by GRASP/Models, while the detailed properties such as AE, AODF and AODC have similarly good validation metric as GRASP/HP. This suggests that the GRASP/Component products provide the overall most consistent both total and detailed aerosol properties (e.g., spectral AODF, AODC, and SSA etc.) with respect to previous GRASP (Models, HP, and Optimized) products.".

3. Figure 2: I guess STD here is calculated from daily data? Anyway, it is better to specify how the authors do the calculation.

**Response:** Thank you for this comment. We added more descriptions and explanations in the revised version (Lines 346-348): "Figure 2 shows the corresponding standard deviations (STD, calculated from all months data during the period 2005－2013, same for other components in the following) divided by the mean of BC concentration."

4. Also, in my opinion, the manuscript would benefit from editing help from someone with full professional proficiency in English.

**Response:** We appreciated the reviewer's conscientious correction very much and we also have checked the grammar and typos carefully and corrected them in the revised manuscript.

**Minor corrections:**

1. Lines 24: "an intermediate retrievals" -> "intermediate retrievals"

**Response:** Done.

2. Line 30: "concentration" -> "concentrations"

**Response:** Done.

3. Line 32: "aerosol" -> "aerosols"

**Response:** Done.

4. Line 45: the full name of IPCC is needed when first mentioned.

**Response:** It was added in the revised version.

5. Line 58: "the importance of having sufficient spectral resolution of measurements to capture the differences" -> "the importance of sufficient spectral resolution of measurements on capturing the differences"

**Response:** Done.

6. Line 85: I feel that "and sometimes the same" is not necessary here.

**Response:** It was removed in the revised version and more explanations were added to make it clear.

7. Lines 101: "has" -> "have"

**Response:** Done.

8. Line 106: "the impact of satellite polarimetry on aerosol monitoring remains fairly." I do not quite understand this sentence. Can the authors rephrase it?

**Response:** It is rephrased and clarified (Lines 115-123): "However, due to rather limited amount of available multi-angle polarization observations and the complexity in their interpretation, the added value of satellite polarimetry on aerosol monitoring remains questionable. Indeed, the polarimetry has enhanced sensitivities to numerous atmospheric parameters and inversion algorithms are required to consider all these sensitivities adequately. Partially due to this complexity, the practical advantages of multi-angular polarimetric retrieval were not convincingly exhibited by the available operational multi-angular polarimetric products in the past for a long time and only recently the advanced aerosol products (including present study) make the advantages

of the polarimetry for aerosol remote sensing more evident (e.g., see discussion in Dubovik et al., 2019, 2021b)."

9. Lines 122-123: the full names of MAP and SRON are missing.

**Response:** We added the full names of MAP and SRON in the revision.

10. Line 147: "This is significant" -> "This is a significant"

**Response:** Done.

11. Line 160: ". This study" -> ", this study"

**Response:** Done.

12. Line 183: "has" -> "have"

**Response:** Done.

13. Line 223: "0.1° and 1°" did the authors mean 0.1° and 0.1°?

**Response:** There are two different spatial resolution for Level 3 products: 0.1° and 1°. We rewrote the sentence in the revision (Line 244) to clarify that as "There are two different spatial resolution of 0.1° and 1° for the Level-3 products".

14. Line 225: It is better to clarify how the authors transform from original resolution to MERRA-2 resolution, by interpolation or aggregation?

**Response:** We added more descriptions and explanations to clarify it in the revision.

15. Line 234: "contribution" -> "contributions"

**Response:** Done.

16. Line 239: "factions" -> "fraction"

**Response:** Done.

17. Line 241: Can authors explain "fractions of 6 components" a bit more? Is it the mass fraction or extinction fraction?

**Response:** We added more descriptions and explanations in the revision (Lines 267-272): "Thus, the main conceptual difference of GRASP/Component from GRASP Optimized and High Precision is the retrieval of volume fractions of six components (black carbon, brown carbon, fine- and coarse-mode non-absorbing insoluble, coarse-mode insoluble absorbing, mainly representing iron oxides in mineral dust, and relative humidity for the host calculation) instead of direct retrieval of the real and imaginary parts of complex refractive index at each wavelength (12 parameters in GRASP Optimized and High Precision)".

18. Line 253: "seem" -> "seems"

**Response:** Done.

19. Line 277: "and considering the apparent of aerosol composition climatological patterns." this sentence is difficult to read with grammar error (apparent of). Can the authors rephrase it?

**Response:** Sorry for this mistake, the revised version is (Lines 308-310): "Therefore, the main focus of this study is on the analysis/verifications of aerosol composition (fractions) retrievals and considering the apparent climatological patterns of aerosol composition. ".

20. Line 310: "columnar" -> "column"

**Response:** Done, also through the whole manuscript.

21. Line 619: "Moreover, possible the hygroscopicity of aged dust." Can the authors rephrase this sentence to a more readable format?

**Response:** Thank you for your attention, the revised version is (Lines 672-680): "The aerosol water presence over ocean during the dust transport and near southern Africa during the biomass burning is also rather logical because of higher atmospheric

relative humidity. Moreover, a significant increase of mineral dust hygroscopicity was attributed in several studies to the aging processes and the dust mixing with soluble hygroscopic material (Sullivan et al., 2009; Tang et al., 2016). Impact of such mixtures on remote sensing observations was also observed (Derimian et al., 2017; Falkovich et al., 2004)."

**Reviewer #2**:

This study presents the dataset for the climatology of aerosol components globally, which was derived by GRASP (Generalized Retrieval of Atmosphere and Surface Properties algorithm) component approach from POLDER-3 satellite observations. The spatial and temporal distributions of satellite-based aerosol components on a global scale are discussed and show reasonable agreement with general expectation, as well as the comparisons of BC and Dust between GRASP component retrievals and MERRA-2 products. The dataset of aerosol components presented in this study provides additional inside information about aerosol properties that is much difficult to be obtained but imperative information for improving the estimation of chemical transport models. The dataset and its more applications in future will be interesting for the scientific community. The manuscript is well-written and well-organized. Therefore, I have some minor comments before it could be accepted for publication.

**Response:** Thank you very much for the time and efforts you have put into reviewing the manuscript. We are very grateful for your positive evaluations and helpful comments on our work, which have enabled us to improve the manuscript. Here are our point-by-point responses to the comments:

**Minor comments:**

1. The full name of abbreviations should be provided when it is mentioned at the first time. Such as, POLDER-3 in Line 21, AERONET in Line 34, IPCC in Line 45 etc. Please check it in the whole manuscript.

**Response:** We thank the reviewer for the attention and pointing this out. We added these abbreviation meanings in the revised version, as well as for AE, AODC, AAOD, and SSA in lines 176-177.

2. Line 75: "Ganguly et al. (2009b, 2009a)" should be "Ganguly et al., (2009a, 2009b)"

**Response:** Done.

3. Line 99: "look-up table (LUT)" should be "Look-Up Table (LUT)"

**Response:** Done.

4. Lines 145-147: Please add references to "This GRASP/Component approach derives fractions of aerosol components together with size distribution and non-spherical fraction of aerosol particle directly from the measured radiances without an intermediate step of optical aerosol properties retrieval."

**Response:** Done.

5. Line 166: Please add references to "based on preliminary sparse analysis of derived aerosol component information". Please clarify what you could obtain from the study of Zhang et al. (2021b) in Line 167.

**Response:** We added references and more descriptions in the revision (Lines 185-187): "based on preliminary sparse analysis of derived aerosol component information (Li et al., 2019, 2020a, 2020b) and on the results of study by Zhang et al. (2021b) demonstrating component approach can provide comparable and sometime even better aerosol optical products".

6. Line 273: "(Li et al., 2020b, 2020a; Zhang et al., 2021)" should be "(Li et al., 2020a, 2020b; Zhang et al., 2021)"

**Response:** Done.

7. Line 350: The reference of "(Schuster et al., 2016b)" in Line 350 is same to the reference of "Schuster et al. (2016)" in Line 63. Please modify and remove one from the reference list.

**Response:** Done.

8. Please clarify what 6 components are in the Line 244 and "the retrieval of fractions of 6 components" should be "the retrieval of 6 components fractions"

**Response:** We added more descriptions and explanations in the revised version (Lines 267-272): "Thus, the main conceptual difference of GRASP/Component from GRASP Optimized and High Precision is the retrieval of volume fractions of six components (black carbon, brown carbon, fine- and coarse-mode non-absorbing insoluble, coarse-mode insoluble absorbing, mainly representing iron oxides in mineral dust, and relative humidity for the host calculation) instead of direct retrieval of the real and imaginary parts of complex refractive index at each wavelength (12 parameters in GRASP Optimized and High Precision)".

9. Lines 263-266: "Thus, using fractions of the components and relative humidity as variable parameters, the refractive index of a particle composed by several insoluble components (e.g., BC, BrC, mineral dust etc.) suspended in such host were determined by the MG equations based on the calculation of electric fields". I do not understand this sentence. Please clarify.

**Response:** The sentence is rephrased and clarified as follows (Lines 294-298): "First, the complex dielectric constant of each aerosol component is computed from the corresponding complex refractive index. Then, the Maxwell Garnett dielectric functions are used for calculating the dielectric constant of the aerosol mixture from component fractions and their dielectric constants. At last, the complex refractive index of aerosol mixture can be obtained from the complex dielectric constant of aerosol mixture.".

**Reviewer #3**:

"Climatology of aerosol components concentration derived by GRASP algorithm from multi-angular polarimetric POLDER-3 observations" submitted to ESSD is a well written paper that studied the global climatology of aerosol species based on GRASP-Component algorithm on POLDER-3. The paper describes the GRASP-Component algorithm, highlights its advantage in retrieving aerosol component, presents the aerosol component global climatology, and compared it to MERRA-2 reanalysis. There are many interpolations of the GRASP results and explanation of potential error sources. The paper is well organized, and the presented climatology can be helpful in terms of understanding the global aerosol type distribution and changes. My main issue first is it requires more discussion of uncertainties in GRASP-Component algorithm and similarly some of the typing climatology. Second is that it needs some more discussion of how this community can use this information, especially in terms of bridging the gaps between remote sensing and modeling communities, which I think is the real advantage of this product.

**Response:** Thank you very much for the time and efforts you have put into the thoughtful reading and reviewing the manuscript. We are very grateful for your constructive evaluations and helpful comments on our work, which have enabled us to improve the manuscript.

First, the detailed uncertainties associated with these refractive indices of all kinds of components had been estimated in a previous study of Li et al. (2019), as well as the impacts of aerosol loading, we now emphasize this in the revised version. That is, we now provide general descriptions and information about the uncertainties of the component retrievals in the corresponding results section (in addition to what has been provided and discussed in the original version of the manuscript). For example, in lines of 385-390 for BC uncertainty: "In this respect it can be noted that based on the sensitivity tests and uncertainty assessments in the study of Li et al. (2019), the uncertainty in BC fraction is within 50% when AOD (440 nm) is larger than 0.4 and fraction is higher than 0.01. This BC uncertainty is mainly resulting from the reported in the literature highly variable complex refractive index of BC, e.g., from $1.75 + 0.63i$

to 1.95 + 0.79i (Bond et al., 2013; Bond and Bergstrom, 2006). Similarly, the uncertainty in BC retrievals from ground-based AERONET measurements is about 50% (Schuster et al., 2016).", in lines of 442-446 for BrC uncertainty: "For the proper data interpretation, it is worth noting that the uncertainty in BrC fraction is normally less than 50% if the BrC fraction is above 0.1, even for very low AOD (smaller than 0.05), although the uncertainty is large in the case of small BrC fraction and low aerosol loading; the error estimations for GRASP/Component were conducted in (Li et al., 2019)." , in lines of 513-518 for CAI uncertainty: "Based on the results of Li et al. (2019), the uncertainty in CAI fraction associated with employed refractive index is within 50% excluding the case of very low CAI fraction, below 0.005. Thus, the CAI concentrations are expected to have large uncertainties over ocean and in high latitude near the polar region (such as the ocean around 60◦ S) where also the AOD is generally very low. Also, cloud contaminations that are more probable in those high latitude areas can be misinterpreted as apparent dust like aerosols." and so on.

We added more descriptions and discussions in the abstract and conclusions to highlight the potential implications of bridging the gaps between aerosol component products and model estimations in the future study. Lines 43-45 in the abstract of revised version: "The extensive satellite-based aerosol component dataset is expected to be useful for improving global aerosol emissions and component-resolved radiative forcing estimations." In addition, lines 807-821 in the conclusions of the revised version: "More importantly, the data assimilation of this extensive satellite-based aerosol component dataset can importantly contribute to improving global aerosol emissions estimation and further improvement of accuracy of the estimated aerosol radiative forcing in general and per aerosol component in particular. For instance, the AOD and the AAOD products derived from POLDER-3 observations have been already used to constrain GEOS-Chem inverse modeling for the improvement of global black carbon, organic carbon and desert dust aerosol emissions (Chen et al., 2018, 2019). Using the presented in this study additional satellite-based aerosol component products, the further improvement of global aerosol emissions estimation is thus expected. The presented efforts are also in line with the studies suggesting that the

employment of satellite-constrained anthropogenic and natural aerosol emissions by the climate models is required to improve the accuracy of aerosol radiative forcing estimations (e.g., Bellouin et al., 2020; Quaas et al., 2022). That is, the linear trends in column concentration of the main aerosol components, such as BC, BrC, CAI and CNAI as shown in Fig. S1 in the supplement, can provide a better global scale satellite-measured constraints on the properties of anthropogenic (BC and BrC) and natural (CAI and CNAI) aerosols and will contribute to improving the accuracy of anthropogenic aerosol radiative forcing estimations."

Below please find our point-by-point responses to the comments:

1. When in the beginning defining these aerosol components, it will be intuitive to relate these with commonly defined nature aerosol species.

**Response:** We appreciate the referee's suggestion. The definitions of aerosol components are based on a series of sensitivity tests that conducted to develop the aerosol component algorithm. Our component approach bridges directly the quantities of aerosol components as close as possible to that used in the chemical transport models. However, we note that the GRASP/Component approach is only possible if 1) there is significant instrument sensitivity to the parameters that are related to an aerosol component (i.e., complex refractive index), and 2) this sensitivity is maintained while other parameters like the size distribution are adjusted. We need to balance the requirements of chemical transport models and the practical ability of deriving aerosol components from remote sensing measurements. All the mentioned above can make a direct attribution to an aerosol species a bit complex, nevertheless, a table describing the definitions of these components and linking to aerosol species nature was added in supplement of the revised version (Lines 859-864):

**Supplement**

Table S1. The definitions and descriptions of aerosol components with their complex refractive indices at 440 nm and 865 nm.

| Component | Complex refractive index | | Reference |
|---|---|---|---|
| | 0.440 $\mu m$ | 0.865 $\mu m$ | |
| BC represents wavelength-independent strong absorption | 1.95+0.79i | 1.95+0.79i | Bond and Bergstrom (2006) |
| | 1.75+0.63i | 1.75+0.63i | Bond and Bergstrom (2006) |
| BrC represents wavelength-dependent absorption | 1.54+0.07i | 1.54+0.003i | Sun et al. (2007) |
| | 1.54+0.06i | 1.54+0.0005i | Kirchstetter et al. (2004) |
| CAI mainly represents iron oxides contained in the coarse-mode dust particles | 2.90+0.345i | 2.75+0.003i | Longtin et al. (1988) |
| | 2.88+0.987i | 2.72+0.140i | Triaud (2005) |
| CNAI mainly represents coarse-mode non-absorbing dust particles | 1.54+0.0005i | 1.52+0.0005i | Ghosh (1999) |
| | 1.53+0.005i | 1.53+0.005i | Ghosh (1999); Sokolik and Toon (1999); Journet et al. (2014) |
| FNAI represents fine-mode non-absorbing insoluble dust and organic carbon | 1.54+0.0005i | 1.52+0.0005i | Ghosh (1999) |
| | 1.53+0.005i | 1.53+0.005i | Ghosh (1999); Sokolik and Toon (1999); Journet et al. (2014) |
| FNAS represents fine-mode inorganic salts | 1.337+10$^{-9}$i | 1.339+10$^{-8}$i | Tang et al. (1981); Gosse et al. (1997) |
| FAWC represents fine-mode aerosol water content | 1.337+10$^{-9}$i | 1.329+10$^{-6.5}$i | Hale and Querry (1973) |

**References**

Bond, T. C. and Bergstrom, R. W.: Light Absorption by Carbonaceous Particles: An Investigative Review, Aerosol Sci. Technol., 40(1), 27–67, doi:10.1080/02786820500421521, 2006.

Ghosh, G.: Dispersion-equation coefficients for the refractive index and birefringence of calcite and quartz crystals, Opt. Commun., 163, 95–102, doi: 10.1016/S0030-4018(99)00091-7, 1999.

Gosse, S. F., Wang, M., Labrie, D., and Chylek, P.: Imaginary part of the refractive index of sulfates and nitrates in the 0.7-2.6 micron spectral region, Appl. Optics, 36, 3622–3634, 1997.

Hale, G. M. and Querry, M. R.: Optical Constants of Water in the 200-nm to 200-microm Wavelength Region, Appl. Optics, 12, 555–563, doi: 10.1364/AO.12.000555, 1973.

Journet, E., Balkanski, Y., and Harrison, S. P.: A new data set of soil mineralogy for dust-cycle modeling, Atmos. Chem. Phys., 14, 3801–3816, doi:10.5194/acp-14-3801-2014, 2014.

Kirchstetter, T. W., Novakov, T. and Hobbs, P. V.: Evidence that the spectral dependence of light absorption by aerosols is affected by organic carbon, J. Geophys. Res. Atmos., 109(D21208), doi:10.1029/2004JD004999, 2004.

Longtin, D. R., Shettle, E. P., Hummel, J. R., and Pryce, J. D.: A Wind Dependent Desert Aerosol Dust Model: Radiative Properties, Scientific Report No.6, 1988.

Sokolik, I. N. and Toon, O. B.: Incorporation of mineralogical composition into models of the radiative properties of mineral aerosol from UV to IR wavelengths, J. Geophys. Res.-Atmos., 104, 9423–9444, doi:10.1029/1998JD200048, 1999.

Sun, H., Biedermann, L., and Bond, T. C.: Color of brown carbon: A model for ultraviolet and visible light absorption by organic carbon aerosol, Geophys. Res. Lett., 34, L17813, doi: 10.1029/2007GL029797, 2007.

Tang, I. N., Wong, W. T., and Munkelwitz, H. R.: The relative importance of atmospheric sulfates and nitrates in visibility reduction, Atmos. Environ., 15, 2463–2471, 1981.

Triaud, A. H. M. J.: Earth observation data group: aerosol refractive index archive, available at: http://eodg.atm.ox.ac.uk/ARIA/data?Minerals/Hematite/(Triaud_2005)/hematite_Triaud_2005.ri (last access: 28 October 2019), 2005.

2. Both in BC and BrC annual maps, I couldn't find elevated signature of smoke over Southeast Asia. But we know that that is one of the major regions that smoldering burring occurs with most of the dry year during El nino phase. Is it due to large cloud coverage? If so, can data availability also line contoured on top of the global map?

**Response:** Thanks a lot for this comment. Yes, indeed there are strong emissions of BC and BrC attributed to the biomass burning events during the MAM (March, April, and May) seasons. In fact, the GRASP/Component retrievals are able to characterize well the temporal and spatial variations of light-absorbing carbon (BC and BrC) in

East and South Asia (such as in Figure R1) and it was discussed in detail by Li et al. (2020). However, this regional effect is indeed not visible in global scale averaging, we therefore added a text to clarify and emphasize this phenomena in the revised version as follows (Lines 366-373): "We should point out that many BC particles are generated from anthropogenic activities such as in China and India (shown in Fig. 4), however, our global climatology also indicates that the BC concentration emitted from biomass burning in Africa and South America produces much higher BC concentrations than the anthropogenic emissions, e.g., in China and India (see Fig. 1 and 3). The climatology of the BC mass concentrations over East and South Asia regions are studied in detail by Li et al. (2020b), which demonstrated that GRASP/Component retrievals can characterize well the temporal and spatial variations of light-absorbing carbon (BC and BrC) in East and South Asia."

[Figure]

Figure R1. The BC column mass concentration (mg/m$^2$) for MAM season over East and South Asia during the period 2005 to 2013.

Reference:

Li, L., Che, H., Derimian, Y., Dubovik, O., Schuster, G. L., Chen, C., Li, Q., Wang, Y., Guo, B. and Zhang, X.: Retrievals of fine mode light-absorbing carbonaceous

aerosols from POLDER/PARASOL observations over East and South Asia, Remote Sens. Environ., 247, 111913, doi:10.1016/j.rse.2020.111913, 2020b.

3. In CNAI map Figure 13, we see a belt of coarse non-absorbing insoluble aerosol near central-south Africa. That is one region that burning of savanna will occur, but very limited dust shall be found there. Similar region was highlighted up in BC map, which shows that these are burning region as well. The cause of these signal needs to be explained.

**Response:** Thank you for pointing this out. This comment is related to the definition of CNAI in Table S1 and to the reply to the Comment #1. Besides the instrument sensitivity question, the components distinguishing is a challenge not only for the proposed GRASP/Component approach, but also for any other efforts aimed to retrieve aerosol component concentration from remote sensing measurements. In the current GRASP/Component retrievals, CNAI (coarse-mode non-absorbing insoluble component) mainly represents coarse-mode non-absorbing dust particles, in particular over the desert dust regions. However, the reason for a more general name "CNAI" used in the definition is that the non-absorbing dust and organic aerosol have close refractive indices. Thus, the GRASP/Component retrievals indicate CNAI in biomass burning regions due to possible presence of coarse-mode organic particles emitted during the biomass burning with strong carbonaceous aerosol emissions, in addition it can be a signature of aged biomass burning aerosols. We added more descriptions and explanations in the revision (Lines 587-589): "We should also point out that some CNAI particles observed near central-south Africa might be attributed to the coarse-mode non-absorbing insoluble organic particles that probably exist when and where there are strong carbonaceous aerosol emissions during the biomass burning events (in Fig. 13)".

4. The explanation of FNAS map in Figure 17 mentioned aged dust, however, it is hard to believe that aged dust will occur over land. Thus, it is unclear to me what is causing FNAS signal over Africa during DJF, JJA, and SON. To me these looks like

signals from biomass burning again, but is burning produce fine mode non-absorbing soluble aerosols?

**Response:** Yes, we agree that FNAS retrievals over Africa during the DJF, JJA, and SON seasons are associated with the fine-mode non-absorbing soluble carbonaceous component generated from the biomass burning events. Indeed, we also provided such explanation in the original manuscript (Lines 600-602 in the original manuscript): "However, the NAS also can be aged hydroscopic components such as water-soluble organic carbon generated from biomass burning or anthropogenic activities, and water-soluble mineral ions provided by dust particles.". Also in lines 606-609 in the original manuscript: "The non-absorbing water-soluble organic carbon, that can be emitted during biomass burning events, can be also detected and interpreted as FNAS, e.g., as it occurs in southern Africa.". The aged dust we mentioned in this part is for the discussions on the FWAC retrievals near or over ocean. Nevertheless, we added more descriptions and explanations in the revised version (Lines 677-680) to make it clear: "Moreover, a significant increase of mineral dust hygroscopicity was attributed in several studies to the aging processes and the dust mixing with soluble hygroscopic material (Sullivan et al., 2009; Tang et al., 2016). Impact of such mixtures on remote sensing observations was also observed (Derimian et al., 2017; Falkovich et al., 2004)."

5. MERRA2 has more BC than GRASP-C globally, especially over East Asia. First of all, the recent GOCART model included BrC as well. (G.P. Schill, K.D. Froyd, H. Bian, A. Kupc, C. Williamson, C.B. Brock, E. Ray, R.S. Hornbrook, A.J. Hills6, E.C. Apel, M. Chen, P. Colarco, and D.M. Murphy, The ubiquity of dilute, aged smoke in the global remote troposphere and its effect on climate, Nature Geoscience, 13(6), doi:10.1038/s41561-020-0586-1, Jun., 2020.) Author can use that version to compare BC and BrC separately. Second, it does seem like some of the biomass burning signal is shown in dust related component (see what I pointed out before.)

**Response:** Thanks a lot for the updated information about the GOCART BrC estimation. However, the data of GOCART BrC is not yet public. We cannot identify

an available website with open accessing and the data availability. In addition, MERRA-2 reanalysis data has been validated and evaluated by many studies and also been applied extensively to investigate the global or regional aerosol optical properties. Therefore, in the current study we show the comparisons of MERRA-2 products and GRASP retrievals. At the same time, the next sentence together with this reference were added (Lines 760-761): "The comparisons of GOCART BrC estimations (Schill et al., 2020) and GRASP BrC retrievals are expected to be done in a future study." In addition, the second part of this comment is indeed getting back to the Comment #3.

Reference:

Schill, G. P., Froyd, K. D., Bian, H., Kupc, A., Williamson, C., Brock, C. A., Ray, E., Hornbrook, R. S., Hills, A. J., Apel, E. C., Chin, M., Colarco, P. R. and Murphy, D. M.: Widespread biomass buring smoke throughout the remote troposphere, 6, 422-427, doi: 10.1038/s41561-020-0586-1, 2020.

6. MERRA2 has a lot more dust over dust belt. I agree that if GRASP-C produced dust cannot be represented only by one component, the systematic low at North Africa is expected. But the high value from GRASP-C product over southern Africa needs to be explained.

**Response:** This comment is related to the Comment #3. In addition to the response to the Comment #3, the next sentence was also added in the revised manuscript (Lines 774-776): "We note that GRASP CNAI retrieval shows slightly higher than MERRA-2 dust in the southern Africa, which might be related to the probably existing coarse-mode organic particles during the biomass burning seasons with strong carbonaceous aerosol emissions".

7. Line 236. How are these refractive indexes determined for components? Especially for mineral dust, large variation can occur based on the origin of the dust. Summarize the approach and uncertainties associated with it.

**Response:** The first part of this comment is indeed related to Table S1 in the Comment #1. The detailed uncertainties associated with these refractive indices of all kinds of components have been estimated in the study of Li et al. (2019). In this manuscript, we provided general descriptions and information about the uncertainties of the component retrievals in the corresponding results section. For example, in lines of 385-390 for BC uncertainty: "In this respect it can be noted that based on the sensitivity tests and uncertainty assessments in the study of Li et al. (2019), the uncertainty in BC fraction is within 50% when AOD (440 nm) is larger than 0.4 and fraction is higher than 0.01. This BC uncertainty is mainly resulting from the reported in the literature highly variable complex refractive index of BC, e.g., from 1.75 + 0.63i to 1.95 + 0.79i (Bond et al., 2013; Bond and Bergstrom, 2006). Similarly, the uncertainty in BC retrievals from ground-based AERONET measurements is about 50% (Schuster et al., 2016).", in lines of 442-446 for BrC uncertainty: "For the proper data interpretation, it is worth noting that the uncertainty in BrC fraction is normally less than 50% if the BrC fraction is above 0.1, even for very low AOD (smaller than 0.05), although the uncertainty is large in the case of small BrC fraction and low aerosol loading; the error estimations for GRASP/Component were conducted in (Li et al., 2019)." , in lines of 513-518 for CAI uncertainty: "Based on the results of Li et al. (2019), the uncertainty in CAI fraction associated with employed refractive index is within 50% excluding the case of very low CAI fraction, below 0.005. Thus, the CAI concentrations are expected to have large uncertainties over ocean and in high latitude near the polar region (such as the ocean around 60∘ S) where also the AOD is generally very low. Also, cloud contaminations that are more probable in those high latitude areas can be misinterpreted as apparent dust like aerosols." and so on.

**Other minor stuff**

1. Line 439-444 mentioned both volume ratio and mass ratio. It is confusing for reader to do the conversion, so providing a mass ratio range from the used volume fraction might be easier for reader to understand author's point.

**Response:** Yes, we agree. We condensed the descriptions to discuss the mass ratio in the revision (Lines 486-488): "The maximum mass ratio of absorbing dust to scattering dust in our retrievals was found of about 5%, which is consistent with the known mass fraction of iron oxides varying from 3% to 5% in the desert dust measurements (Guieu et al., 2002; Zhang et al., 2003)."

2. Line 389-390. The statement of BC and BrC relation with smoke is not entirely accurate. Many papers discuss the more absorbing vs. less absorbing smoke, including those from AERONET groups (Tom Eck), the recent one is here (Junghenn Noyes KT, Kahn RA, Limbacher JA, Li Z. Canadian and Alaskan wildfire smoke particle properties, their evolution, and controlling factors, from satellite observations. Atmospheric Chemistry and Physics Discussions. 2021 Nov 3:1-34.)

**Response:** We included this reference to provide additional descriptions and explanations in the revised version (Lines 432-434): "Although statistically significant differences of smoke properties have been observed for fires in different fuel types such as forests, savannas, and grasslands (Noyes et al., 2021), our component retrievals indicate that …".

Reference:

Noyes, K. T. J., Kahn, R. A., Limbacher, J. A., and Li, Z.: Canadian and Alaskan wildfire smoke particle properties, their evolution, and controlling factors, from satellite observations, Atmos. Chem. Phys. Discuss., 1–34, doi: 10.5194/acp-2021-863, 2021.

3. Why are standard deviation plots using 4 identical color bars while seasonal map using one unified color bar? Can they be consolidated?

**Response:** We replotted and replaced the figures in the revision.

4. Color bar in seasonal map should have units on it.

**Response:** We replotted and replaced the figures in the revision.

5. Line 325-326 "that low…is small." This sentence is confusing. Low STD/MEAN is observed when intensity of emission is small? or low STD/MEAN is observed due to low emission. Nevertheless, in MAM the STD of BC is high in Asia, indicating changing of BC emissions.

**Response:** The sentence was rewritten in the revision (Lines 361-363): "Fig. 2 also indicates high STD/Mean values for BC concentration in Asia during the MAM season, which is associated with large interannual variability of fires in the biomass burning season (such as over Indo-China Peninsula)."

6. Line 279 "minimal bias", be specific.

**Response:** The value is provided.

7. Line 96. Flower has some volcanic paper with Ralph Kahn.

**Response:** Thank you for your suggestions, several additional references are included in the revision.

8. Line 83. Data assimilation paper adding these two citations (Zhang J, Reid JS, Westphal DL, Baker NL, Hyer EJ. A system for operational aerosol optical depth data assimilation over global oceans. Journal of Geophysical Research: Atmospheres. 2008 May 27;113(D10).; Shi Y, Zhang J, Reid JS, Hyer EJ, Hsu NC. Critical evaluation of the MODIS Deep Blue aerosol optical depth product for data assimilation over North Africa. Atmos Measure Tech Discuss. 2012 Oct 24;5(5):7815-65.)

**Response:** Thanks a lot for your suggestions, these references are included now.

9. Line 32, "The aerosol optical properties" means what properties? Need clarification.

**Response:** We added more descriptions and explanations in the revision.